# Genotoxicity of Coffee, Coffee By-Products, and Coffee Bioactive Compounds: Contradictory Evidence from In Vitro Studies

**DOI:** 10.3390/toxics13050409

**Published:** 2025-05-18

**Authors:** Maryam Monazzah, Dirk W. Lachenmeier

**Affiliations:** 1Erasmus Programme, Department of Comparative Biomedicine and Food Science, University of Padua, Viale dell’Università 16, 35020 Legnaro, Padua, Italy; maryam.monazzah@studenti.unipd.it; 2Chemisches und Veterinäruntersuchungsamt (CVUA) Karlsruhe, Weissenburger Strasse 3, 76187 Karlsruhe, Germany

**Keywords:** coffee, coffee by-products, genotoxicity, antimutagenicity, Ames test, comet test, micronucleus test, DNA damage, roasting process, bioactive compounds, food safety

## Abstract

Coffee and coffee by-products, such as coffee cherries, coffee flowers, coffee leaves, green beans, roasted coffee, instant coffee, spent coffee grounds, and silverskin, contain a complex mixture of bioactive compounds that may exhibit both genotoxic and antimutagenic effects. This article evaluates in vitro studies on the genotoxic potential of coffee and coffee by-products, with a focus on different preparation methods, roasting processes, and key chemical constituents. Furthermore, given the growing interest in utilizing coffee by-products for novel food applications, this review sought to identify knowledge gaps regarding their safety. The impact of metabolic activation, particularly the role of enzymatic detoxification and bioactivation, was examined to better understand the effects on genetic material. The findings suggest that while certain compounds in coffee can induce DNA damage under specific conditions, the overall evidence does not indicate a significant genotoxic risk to consumers. However, further studies, particularly in vivo and human studies, appear necessary to ensure the requirements of novel food applications for some coffee by-products.

## 1. Introduction

Coffee is a highly popular and consumed beverage, enjoyed across diverse age groups, regions, and socioeconomic strata. It also holds significant economic importance as the second most traded commodity worldwide and the leading agricultural commodity, sustaining the livelihoods of over 30 million smallholder households [1,2]. Among the various species of coffee, *Coffea arabica* L. (commonly known as Arabica) and *Coffea canephora* L. (commonly known as Robusta) hold significant economic importance, accounting for approximately 75% and 25% of global coffee production, respectively [3]. According to USDA Foreign Agricultural Service, world coffee production for 2024/25 is forecast at 6.9 million bags higher than the previous year, to 174.9 million. Global consumption is expected to rise by 5.1 million bags to 168.1 million [4]. Beyond its widespread use as a daily stimulant, coffee is recognized for its biologically active ingredients and its potential to influence human health positively and negatively [5]. The beverage contains a diverse array of bioactive compounds, including chlorogenic acids, caffeine, diterpenes, phenolics, trigonelline, and melanoidins, making it a focus of extensive scientific investigation (Figure 1). Moreover, the expanding coffee industry generates significant by-products, such as husks, hulls, spent coffee grounds, and silverskin. These by-products present opportunities for valorization across various industries, including cosmetics, pharmaceuticals, agriculture, animal feed, the food sector, biorefinery applications, and biopolymer production [6,7,8]. Utilizing these by-products aligns with sustainable development goals of the UN, yet their reuse also raises concerns regarding human consumption safety, particularly their possible genotoxic and mutagenic potential, and environmental impacts.

European legislation provides a rigorous framework for the validation of novel foods or ingredients, such as many coffee by-products [9], which includes a thorough chemical characterization (encompassing toxicity and health-promoting compounds), proposed applications, in vitro testing, in vivo toxicity evaluations, bioactivity assessments, and clinical trials involving humans [10]. Genotoxicity data are pivotal in the risk assessment of substances, including novel foods and ingredients [10,11]. The European Food Safety Authority (EFSA) recommends a tiered genotoxicity testing strategy for novel foods, beginning with in vitro tests such as the Ames test and micronucleus test, and progressing to in vivo studies only if necessary, to ensure efficiency and minimize animal use [11,12].

Evaluating mutagenic potential is essential due to the severe implications genetic damage can have on human health. This evaluation involves analyzing studies on mutation induction alongside tests exploring other genetic impacts. As outlined in the REACH “Guidance on Information Requirements and Chemical Safety Assessment” [13], mutagenicity refers to heritable genetic changes in cells or organisms, including alterations in genes or chromosomes. Clastogenicity involves structural chromosome changes, while aneugenicity refers to changes in chromosome numbers, leading to aneuploidy. Genotoxicity, a broader term, includes any DNA alterations, not necessarily mutations. Tests for genotoxicity and mutagenicity examine DNA damage and mutations to understand the genetic effects of chemical exposure comprehensively.

Coffee’s chemical complexity means that it may possibly exhibit both genotoxic and protective effects, depending on factors such as preparation methods, roasting levels, and the biological systems in which it is tested [14,15,16,17,18,19]. For instance, while chlorogenic acids and caffeine are widely recognized for their antioxidant and antimutagenic properties, they may also generate reactive oxygen species (ROS) under certain conditions, leading to DNA damage [20]. Similarly, roasting processes that enhance coffee’s flavor and aroma lead to the dehydration and degradation of various compounds, including polyphenols and may produce mutagenic Maillard reaction products, such as acrylamide, 5-hydroxymethylfurfural (HMF), furfuryl alcohol and polycyclic aromatic hydrocarbons [15,21,22,23,24]. The protective effect is observed in both caffeinated and decaffeinated preparations, suggesting that compounds other than caffeine contribute to these properties. The antigenotoxic effects are often attributed to coffee’s antioxidant properties [25,26]. A large body of evidence from various cohort and case-control studies consistently demonstrates a protective effect of coffee consumption against liver cancer. This protective effect appears to be consistent across different populations and demographic groups [27].

This review synthesizes findings from over 100 referenced studies evaluating the in vitro genotoxic and antimutagenic effects of coffee, coffee by-products (green beans, roasted coffee, instant coffee, spent coffee grounds, and silverskin), and associated bioactive compounds, with the goal of determining the overall safety of these materials for human consumption. Of particular interest are the differential effects observed between various coffee preparations and their by-products, as well as the role of key chemical constituents. These studies also include evaluations with and without metabolic activation systems, such as the S9 mix, which simulate the enzymatic processes of the liver. The inclusion of these systems is particularly important as many compounds may undergo bioactivation to form reactive intermediates or detoxification during metabolism, leading to context-dependent effects. Considering the increasing use of coffee by-products in novel food applications, this review further seeks to identify knowledge gaps in the current literature concerning their safety, highlighting the need for further research.

## 2. Materials and Methods

Electronic searches of the literature were conducted using several electronic databases, including PubMed (National Library of Medicine, Bethesda, MD, USA), and Google Scholar (Google LLC, Mountain View, CA, USA). For chlorogenic acid, there was a literature overview available from a previous review of Behne et al. [28].

To expand the search beyond the initial results, a citation tracking approach was employed, tracing both the bibliographic roots of identified publications and the subsequent work of authors in the field via Google Scholar (Google LLC, Mountain View, CA, USA). To provide a comprehensive overview of the research, no date restrictions were placed on the literature search, as many important publications on this topic are older.

A wide range of search terms were used, including coffee (green beans, roasted coffee, instant coffee), *Coffea*, coffee blossom, coffee flowers, coffee cherry, coffee leaf, coffee leaves, cascara from coffee, coffee pulp, coffee husk, mucilage, coffee parchment, silver skin, coffee bean, coffee seed (grounded, roasted), extracted ground coffee residuals, coffee grounds, coffee grounds oil extract, coffee oil, coffee key chemical constituents, genotoxicity, Ames test, comet test, chromosomal aberration tests, and micronucleus test. Only studies employing in vitro test methodologies were included.

## 3. Results

### 3.1. Overview of Genotoxicity Testing Methods

In line with EFSA’s guidance, genotoxicity assessment strategies should be tailored to the specific characteristics of the novel food, taking into account whether it is a complex mixture, contains nanomaterials, or involves microorganisms [11]. The first tier involves a basic battery of in vitro tests including bacterial reverse mutation assay or Ames test (OECD TG 471) [12] and in vitro micronucleus assay (OECD TG 487) [29]. For nanomaterials or proteins/peptides, other tests like mammalian gene mutation assays or the bacterial reverse ‘treat and plate’ method might be more appropriate [30]. If the in vitro micronucleus test is positive, further investigation for aneugenicity is required [11]. For the Ames test, the results are presented as revertant colonies per plate, including controls, with a positive result indicating the induction of point mutations (base substitutions or frameshifts) in strains. A positive result is determined by a concentration-related increase or a reproducible increase at one or more concentrations in the number of revertant colonies per plate in at least one strain, with or without metabolic activation. The decision of a positive response should be based primarily on their biological relevance, not just on statistical significance. In rare cases, the data might not show clearly positive or negative results, leaving the substance’s activity uncertain even after repeating the experiment multiple times [12]. In the in vitro micronucleus (MN) test, evaluation focuses on the frequency of micronucleated binucleated cells, with a positive result requiring a statistically significant increase in micronuclei compared to the negative control, a dose-related increase, and at least one result exceeding the historical negative control data’s 95% control limits, indicating potential chromosome breaks and/or gain or loss. Unclear positive or negative MN test results necessitate expert review and/or further investigation (e.g., scoring more cells, repeating the experiment with modified conditions) [29]. In addition to commonly employed genetic toxicity tests, other established methods include the UMU assay, which evaluates DNA damage based on SOS-induced β-galactosidase activity in *Salmonella typhimurium*; the prophage induction assay, which measures lytic phage release in *Escherichia coli* following an SOS response; and the Ara assay, which detects forward mutations by assessing L-arabinose resistance in *S. typhimurium* [31].

If Tier I tests yield positive or ambiguous results, Tier II in vivo genotoxicity studies are conducted. The Scientific Committee recommends the following as suitable in vivo tests: mammalian erythrocyte micronucleus test (OECD TG 474), transgenic rodent somatic and germ cell gene mutation assays (OECD TG 488), in vivo comet assay (no OECD TG at present; internationally agreed protocols available), in vivo micronucleus test (especially an appropriate follow-up test for in vitro clastogens and aneugen), and in some cases, when more detailed analysis is required about structural chromosome changes, an in vivo mammalian bone marrow chromosome aberration test (OECD TG 475) may be an alternative [10]. Tier III is confirmation of in vivo genotoxicity. If in vivo tests are positive for novel foods, the substance is considered genotoxic in vivo, and no further testing is needed [11].

Following the genotoxicity testing framework recommended by EFSA, numerous studies have applied these standard assays to evaluate the genotoxic potential of coffee and its by-products. A summary of these studies is presented in Table 1 and Appendix A Table A1, and they are reviewed in detail in the following sections.

### 3.2. Studies Showing No Effects

#### 3.2.1. Green Coffee Beans and Unprocessed Coffee Products

Albertini et al. [33] assessed the mutagenicity of green coffee beans using the Ames test with *S. typhimurium* TA100. Lyophilized green coffee bean powder was tested at concentrations ranging from 5 to 50 mg/plate without metabolic activation. The results indicated that green coffee beans did not exhibit mutagenic activity. The results of a study by Kosugi et al. [15] also indicated that green coffee was not mutagenic in the Ames test and did not induce phage activity in *E. coli* GY5027/GY5022. Dorado et al. [34] assessed the mutagenic potential of green coffee using the L-arabinose resistance assay and the Ames test in *S. typhimurium*. The findings indicated that green coffee exhibited no mutagenic activity. Using the comet assay, Glei et al. [35] found no DNA damage or toxicity from green coffee in human liver (HepG2) and colon (HT29) cells. Similar results were reported in non-human mammalian cells. Santa-Maria et al. [36] found that unroasted green coffee did not induce sister chromatid exchanges (SCEs) in Chinese hamster CHO-K1 cells.

Heimbach et al. [32] evaluated the genotoxic potential of whole coffee fruit (including the seed or bean) extract from *Coffea arabica*, using bacterial mutagenicity. Whole powder, water, and ethanol extracts were tested in the Ames assay with *S. typhimurium* and *E. coli*, using both the plate incorporation and the pre-incubation methods. The whole powder extract was non-mutagenic in all strains except TA1537, which showed toxicity at 5000 μg/plate (plate incorporation) and ≥316 μg/plate (pre-incubation). The water extract was non-mutagenic in all conditions. The ethanol extract was non-mutagenic except for TA1535 at 5000 μg/plate.

#### 3.2.2. Processed Coffee Products

Da Silva et al. [39] evaluated the mutagenicity, and genotoxicity, of coffee prepared from *Coffea arabica* beans of varying quality (commercial vs. specialty) and roasting degrees (light, medium, dark) using the Ames test (*S. typhimurium* TA100 and TA98), and Cytokinesis-block Micronucleus Test on HepG2 cells. Dark roast coffee (commercial quality) at the highest concentration (50 mg/mL) showed mutagenic activity in TA98 without metabolic activation but lacked a dose-response relationship, indicating no significant mutagenic potential. Specialty coffee samples (light, medium, and dark roasts) showed no mutagenicity, genotoxicity, or clastogenicity. Cytokinesis-block proliferation index (CBPI) and replicative index (RI) values confirmed normal cell division, with no chromosomal damage or abnormal chromosome numbers observed in HepG2 cells.

The mutagenic potential of Americano coffee from popular international chains was evaluated using the Ames test with *S. typhimurium* strains TA98, TA100, and TA1535. The Americano coffee was made by adding hot water to a shot of espresso, which is typically brewed using approximately 7 g of finely ground coffee. Results indicated no significant mutagenic activity in any of the coffee samples, regardless of the presence or absence of metabolic activation (S9), and no dose-response relationship was observed [40].

Blair and Shibamoto [60] investigated the mutagenic potential of different coffee preparations using the Ames test. Thick coffee syrup, the dichloromethane extract of distillates of brewed coffee, and coffee solid residues—along with their dichloromethane and ethanol extracts—showed no mutagenic activity in the tested assays. This suggests that overheating brewed coffee to create a thick syrup does not produce significant mutagenic compounds. Similarly, the solid residues left after overheating brewed coffee did not exhibit mutagenicity in the Ames test at any dose level.

Aeschbacher and Würzner [53] compared the mutagenicity of instant and regular (“home brew”) coffee using the Ames test with *S. typhimurium* strains TA100, TA98, TA1535, TA1537, and TA1538, at concentrations of 0–50 mg/plate. High concentrations of both coffee types induced a two-to-three-fold increase in revertants above spontaneous levels in TA100, but only at bactericidal concentrations, suggesting potential false positives. No mutagenic effects were observed in the other strains. The observed mutagenicity in TA100 was considered weak, below the Ames test positive threshold (less than twice spontaneous revertants), and occurred at coffee concentrations 100 times higher than recommended. S9 metabolic activation completely abolished the mutagenic effects.

The presence of the mixture of several compounds in coffee raises the possibility of artifacts in the Ames test. To address this, Aeschbacher et al. [54] compared the mutagenicity of instant and fresh-brew coffee using the Ames test at different concentrations and explored various factors that might influence or interfere with the test results. Their findings were in alignment with the results of their previous study [53]. The observed mutagenicity and bactericidal effects were closely linked. In addition, they found that caffeine level, histidine content, autoclaving, and pH had no significant impact on the results. The study concluded that both instant and fresh-brew coffee exhibit weak, metabolically deactivated mutagenicity in TA100, likely due to bactericidal interactions rather than true mutagenicity, with no significant influence from the tested factors.

Similar results were observed in mammalian experimental systems. Bichler et al. [59] examined the effects of coffee consumption on DNA stability in humans by analyzing lymphocyte DNA damage in eight individuals with comet assay before and after drinking 600 mL of coffee daily for five days. This study used metal-filtered coffee, which was prepared using the French Press method. Results showed that coffee consumption did not significantly affect DNA migration under standard comet assay conditions.

Abraham and Stopper [25] and Abraham et al. [55] investigated the genotoxic potential of various coffee preparations (caffeinated and decaffeinated instant coffee, filtered and unfiltered, and boiled coffee) using both the micronucleus test and the comet assay in mouse Lymphoma L5178Y. To prepare boiled coffee, coarse ground roasted coffee was boiled in water and then hot-filtered. They observed no induction of either micronuclei or DNA damage at the tested concentrations.

Duarte et al. [41] demonstrated the absence of genotoxicity of instant coffee in the Ames test with *S. typhimurium* TA100.

Edenharder et al. [57] reported similar results using the comet assay in Chinese hamster lung fibroblasts (V79 rCYP1A2-rSULT1C1), finding no genotoxicity with coffee, though the specific type of coffee was not specified.

#### 3.2.3. Coffee By-Products

Iriondo-DeHond et al. [63] evaluated the cytotoxicity and genotoxicity of coffee silverskin extract (CSE) and chlorogenic acid (CGA) in HepG2 cells using the alkaline comet assay (for genotoxicity). DNA damage (strand breaks and oxidized purines/pyrimidines) was assessed in the comet assay, with Fpg and Endo III enzymes used to detect oxidized purines and pyrimidines, respectively. Neither CSE nor CGA induced cytotoxicity up to 10,000 and 1000 μg/mL, respectively. Similarly, neither CSE nor CGA induced strand breaks or significant oxidative DNA damage at any tested concentration (10–1000 μg/mL). The study concluded that CSE and CGA are non-genotoxic in HepG2 cells, even at relatively high concentrations.

Genotoxicity of spent coffee grounds (SCGs) from Arabica filter and Canephora espresso coffee preparations was assessed using the comet assay. The results showed that neither preparation induced relevant strand breaks or oxidized purines (FPG-sensitive sites) in the cells [65].

Similar results were achieved in a study that investigated the mutagenic and antimutagenic activities of spent coffee and coffee brews (Arabica filter and Canephora espresso) using the Ames test. Under the tested conditions, neither the coffee extracts nor their bioactive compounds exhibited mutagenic or toxic effects. The study indicated that these substances are not inherently genotoxic and instead demonstrate antimutagenic properties, particularly with S9 metabolic activation [62].

Pectin-based composite films incorporating cork and coffee grounds were developed as a sustainable alternative to plastic packaging. Cruz et al. [56] assessed the safety of these films using the Ames test with *S. typhimurium* strains TA98 (for frameshift mutations) and TA100 (for point mutations). Using 0.25 g of coffee grounds in the film, the Ames test showed negative results for both strains under both conditions. This indicates that no DNA-reactive (mutagenic) substances were detected, confirming the safety of the materials.

#### 3.2.4. Coffee Bioactive Compounds

The mutagenic potential of mangiferin, found in *C. arabica* leaves, was assessed using the Ames test. The Ames test showed no biologically relevant increases in revertant colonies in any strain, with or without S9, even at the highest dose (5000 µg/plate), concluding mangiferin is non-mutagenic under these conditions. Based on the negative Ames and in vivo micronucleus results, mangiferin is considered non-genotoxic, and its presence in coffee leaves is considered safe by the EFSA [82].

Glei et al. [35] reported no evidence of DNA damage or toxicity from chlorogenic acid in HepG2 and HT29 cells, as assessed by the comet assay.

Fung et al. [68] evaluated the mutagenic activity of four coffee ingredients—caffeic acid, chlorogenic acid, pyrazine, and trigonelline—using the *Salmonella* plate incorporation assay and the L5178Y TK⁺/^−^ mouse lymphoma assay. In the *Salmonella* assay, both caffeic and chlorogenic acids, as well as pyrazine and trigonelline, were negative for mutagenicity in all tested strains (TA1535, TA1537, TA1538, TA98, and TA100), with and without S9 metabolic activation. However, in the mouse lymphoma assay, only pyrazine and trigonelline were negative in this assay. These results were consistent with previous findings of pyrazine and chlorogenic acid genotoxicity [69,72].

Hernandes et al. [71] investigated the effects of caffeic acid and chlorogenic acid, two major coffee polyphenols, on cytotoxicity and genotoxicity in HL-60 and Jurkat leukemia cell lines using the comet and micronucleus assays. Neither caffeic acid nor chlorogenic acid induced DNA damage in the comet assay or increased micronuclei frequency in the micronucleus assay at the tested concentrations, indicating they are not genotoxic or mutagenic in these cell lines. However, chlorogenic acid at 100 µM induced global DNA hypomethylation in Jurkat cells but not HL-60 cells, suggesting a cell-specific effect. Caffeic acid did not alter global DNA methylation in either cell line.

Several studies showed that chlorogenic acid did not exhibit mutagenicity in the *S. typhimurium* TA98 strain, both with and without S9 metabolic activation [79,80]. The results in Chinese hamster V79-6 cells also confirmed that chlorogenic acid (500 nmol/mL) did not exhibit genotoxicity [81].

Stich et al. [69] also reported that chlorogenic acid, caffeic acid and quinic acid were not mutagenic in *S. typhimurium* strains TA98 and TA100.

Aeschbacher et al. [42] found that pure caffeine, tested at levels equivalent to those in regular coffee, did not induce chromosomal aberrations in human lymphocytes, with or without S-9 metabolic activation. Similar results were obtained in a study using *S. typhimurium* BA13 strain, where caffeine was found to be non-mutagenic [43].

Similarly, Nakasato et al. [44] demonstrated the absence of mutagenic activity of caffeine in cultured Chinese hamster lung (CHL) cells.

Duarte et al. [41] used the Ames test to examine the genotoxic potential of caffeine after nitrosation, considering that coffee contains compounds that can react with nitrite under gastric conditions. Coffee exhibited its highest genotoxic activity at doses up to 145 mg/plate, corresponding to a caffeine content of 0–38 nmol. However, when caffeine was tested after nitrosation at doses up to 47 nmol per plate, it showed no genotoxicity, indicating that caffeine is not a key contributor to the genotoxic effects observed in nitrosated coffee.

Majer et al. [91] investigated the genotoxic effects of coffee diterpenoids, cafestol palmitate and a mix of cafestol and kahweol (C + K), in HepG2 cells. No genotoxic effects were observed in the micronucleus assay across a broad dose range.

Aeschbacher et al. [90] assessed the mutagenic potential of coffee aroma constituents using the Ames test. Furans and sulfur-containing compounds showed no mutagenic activity. Certain *N*-heterocyclics, including pyridine, pyrazine, 2,5-dimethylpyrazine, and ethylpyrazine, as well as maleic anhydride, induced slight but non-significant increases in TA102 (+S9). Additionally, 2-methylbutanal and 3-methylbutanal slightly increased mutation frequency in TA98 (+S9) and TA100 (+/−S9), though the effect was not statistically significant.

#### 3.2.5. Compounds Formed During the Maillard Reaction

Janzowski et al. [93] investigated the genotoxicity of HMF using the comet assay for mammalian DNA damage. No DNA damage was detected in V79, Caco-2, primary rat hepatocytes, or primary human colon cells up to the cytotoxic threshold using the comet assay.

Aeschbacher et al. [97] also found that dimethylpyrazine and HMF did not increase revertants in *Salmonella* TA100 and TA98 in the direct plating assay, with or without S9 metabolic activation, at concentrations up to bactericidal levels.

While HMF is generally mutagenic after being metabolized by sulfotransferases (SULTs) from rats or humans [99,100,101], it did not induce micronuclei in HepG2 cells expressing SULTs in the in vitro micronucleus test [94]. In addition, HMF showed no mutagenic activity in the Ames test [94].

### 3.3. Studies Showing Genotoxic Effects

#### 3.3.1. Roasted Coffee

Mutagenic activity has been detected in various heat-treated foodstuffs, including coffee. Research on roasted coffee revealed that mutagenic compounds, formed during roasting, exhibit direct-acting mutagenicity in *S. typhimurium* TA100 [33]. The study examined how roasting time and temperature affect mutagenic compound formation and compared drum roasting (500–600 °C) method with the air-bed roasting (260–330 °C) method. Roasting increased mutagenicity up to 4 min in a drum roaster, with similar results using the air-bed roasting method, indicating the roasting process rather than the method, is critical. Mutagenic compounds were formed even at temperatures below 220 °C, highlighting their rapid formation and heat stability. Aqueous extracts of roasted coffee induced 6–10 times more revertants at 20 mg/plate compared to controls, although higher concentrations reduced mutagenicity due to bactericidal effects. Instant coffee samples served as an internal standard to assess result reproducibility. Its mutagenic activity was comparable to coffee extracts. The findings were consistent across various Arabica and Canephora coffee varieties, emphasizing that mutagens are heat-stable and formed early in roasting, which poses challenges for mitigation without altering coffee quality.

Farag et al. [37] compared the genotoxic effects of roasting and irradiation on coffee green beans, using the MN test on blood samples from six healthy volunteers, and found that roasting at 200 °C (10 min for light coffee, 20 min for dark coffee) significantly increased genotoxicity, with a higher binucleated/micronucleated (BN/MN) ratio observed for longer roasting durations. In contrast, irradiation at 5.0 and 10.0 kGy resulted in lower genotoxicity than roasting, showing a milder effect even at higher doses. A dose-dependent relationship between processing methods and genotoxicity was identified in this study.

Nagao et al. [38] evaluated the mutagenic potential of roasted and instant coffee using the Ames test with *S. typhimurium* strains TA100 and TA98, both with and without metabolic activation (S9 mix). The results showed that both roasted and instant coffee (caffeinated and decaffeinated) exhibited mutagenic activity in TA100 under conditions without S9, inducing 1.4 × 10^5^ to 4.6 × 10^5^ revertants per cup for regular coffee and 5.4 × 10^4^ to 5.8 × 10^4^ revertants per cup for instant coffee. The addition of S9 mix significantly reduced mutagenicity, likely due to enzymatic inactivation or reactions with proteins in S9 [38]. Hemoproteins and catalase activity were found in a partially purified, mutagen-inactivating fraction (cytosol fraction of rat liver) [83]. The study emphasizes that observed mutagenicity does not necessarily imply carcinogenicity. Further research is needed to identify the chemical structures and potencies of the mutagens in coffee that act directly on TA100 [38].

Another publication highlighted the role of the roasting process in generating mutagenic compounds and the impact of DNA repair mechanisms on their effects [15]. The authors assessed the genotoxic potential of roasted, and instant (caffeinated and decaffeinated) coffee using the Ames test (*S. typhimurium* TA100, TA98, TA1535, TA92, *E. coli* WP2 uvrA/pKM101) and prophage induction in *E. coli* GY5027/GY5022. Roasted coffee was mutagenic in the Ames test to *S. typhimurium* TA100 and *E. coli* WP2 uvrA/pKM101 (plasmid-dependent), with weaker activity in TA98. Despite having pKM101 plasmid, which is known to increase sensitivity to certain mutagens, mutagenicity was not observed in *Salmonella* TA92, suggesting that plasmid presence alone is not the sole determinant of sensitivity. S9 metabolic activation abolished this mutagenicity. In the prophage induction assay, brewed and both types of instant coffee induced prophage λ in the repair-deficient GY5027 strain (but not the repair-competent GY5022), implying that DNA lesions caused by coffee are repairable by uvr gene products. Roasted coffee induced phage activity and significantly increased the number of plaques. One cup of coffee (2 g instant) was estimated to have prophage-inducing activity equivalent to 800 µg *N*-methyl-*N*′-nitro-*N*-nitrosoguanidine (MNNG, a known mutagen) and mutagenic activity equivalent to 8 µg MNNG [15].

Blair and Shibamoto [60] investigated the mutagenic potential of dichloromethane extracts of distillates from overheated brewed coffee using the Ames test. While extracts of the distillates obtained from normally heated brewed coffee showed no mutagenicity, extracts of distillates from overheated (150–300 °C) brewed coffee produced mutagenic compounds detectable in *S. typhimurium* strain TA98 with S-9 activation. The findings indicated that overheating coffee may create mutagens, suggesting temperature control is important during preparation.

Dorado et al. [34] used the L-arabinose resistance and Ames test in *S. typhimurium* to investigate the mutagenic potential of roasted and un-roasted coffee beans, ground coffee and instant coffee. Results showed that roasted coffee, but not green coffee, was highly mutagenic, confirming that roasting produces mutagenic compounds. Regular and sugar-roasted coffees had similar mutagenic activity, but instant coffee showed nearly twice the mutagenicity of non-instant coffee. However, when considering mutagenic activity per cup, instant and non-instant coffee exhibited similar levels of mutagenicity. The BA13 strain was the most responsive to coffee mutagenicity, while TA102 had the lowest response. The study highlighted that the L-arabinose resistance (Ara) forward mutation assay detected coffee mutagenicity more effectively than the Ames reversion test. Notably, the mutagenicity of coffee decreased in the presence of the S9 metabolic activation system, indicating that some mutagenic compounds may be detoxified in the body. The study concluded that although several coffee components (including H_2_O_2_, methylglyoxal, glyoxal, and caffeic acid) are mutagenic in the Ara assay, they do not explain coffee’s total mutagenic activity in this assay.

The genotoxicity of *Coffea arabica* coffee, varying in bean quality (commercial vs. specialty) and roasting level (light, medium, dark), was evaluated by Da Silva et al. [39], using the Ames test with *S. typhimurium* TA100 and TA98. Of all samples tested, only the commercial medium roast coffee exhibited genotoxicity in TA100 with S9, suggesting metabolic activation enhances cytotoxic effects.

Bichler et al. [59] investigated the effects of coffee consumption (metal-filtered coffee) on human DNA stability by analyzing lymphocyte DNA damage using the comet assay. Results indicated that DNA migration increased significantly at ≥50 µL/mL, indicating potential genotoxic effects at higher concentrations.

#### 3.3.2. Instant Coffee

Aeschbacher et al. [42] found that instant and home-brewed coffee (both caffeinated and decaffeinated) and coffee aroma increased chromosomal aberrations in human lymphocytes, particularly without metabolic activation (S-9 mix). The mutagenic activity of coffee and coffee aroma was significantly lower in the presence of S-9.

The genotoxic potential of different commercial coffee types (roasted, high roast, blend grounds, and instant) was assessed by measuring sister chromatid exchanges (SCEs) in Chinese hamster CHO-K1 [36]. Results showed that caffeinated instant coffee exhibited the highest genotoxic activity, particularly in the absence of metabolic activation (S-9 mix), suggesting that some mutagens in coffee do not require metabolic activation. Decaffeinated coffees had lower genotoxicity, and green coffee (unroasted) showed no significant effects, implying that roasting and caffeine content may contribute to genotoxic compound formation. Blend coffees (50% roasted and 50% high roast coffee) increased SCEs more than roasted coffees. No significant difference in SCE frequency between blend and high-roast coffees suggested that roasting beyond a certain point does not further increase genotoxicity.

Similarly, in a non-mammalian experimental system, Suwa et al. [45] showed that three types of coffee, freshly brewed coffee, regular instant coffee, and decaffeinated instant coffee are mutagen on *S. typhimurium* TA100, and weakly mutagen on TA98 without S9. In addition, instant coffee and caffeine-free instant coffee showed phage-inducing activities in *E. coli* K12. Sulfite and bisulfite (≤300 ppm) effectively inactivated coffee’s mutagenicity, unlike other reducing agents such as reduced glutathione. While the mechanism of sulfite/bisulfite inactivation is unknown, it was shown to inactivate the mutagenicity of diacetyl and glyoxal, compounds present in coffee.

Johansson et al. [46] investigated the mutagenicity of coffee substitute (a mixture of instant coffee and natural flavor extracted from chicory) and instant coffees using the Ames test. Coffee substitute and instant coffees exhibited mutagenic activity in *Salmonella* strains TA98, YG1024, and YG1029 (the latter strains overexpress *N*-acetyltransferase, which catalyses the activation of heterocyclic aromatic amines), but only with metabolic activation. The observed mutagenicity in *Salmonella* strains, particularly the stronger response in YG1024, suggests the presence of frame-shift mutagens, likely aromatic amines or nitro compounds. YG1024′s enhanced response indicates a need for *O*-acetyltransferase activity for metabolic activation, a characteristic of these types of compounds. The requirement for metabolic activation (S9) further supports the presence of amines, which require enzymatic conversion to become mutagenic.

Shane et al. [18] comprehensively evaluated the mutagenicity of nine coffee formulations (four caffeinated instant, three caffeinated drip, and two decaffeinated) using the Ames test. Drip coffee was made by passing 250 mL of heated water through 25 g of coffee grounds in a filter. All nine formulations exhibited direct-acting mutagenicity in all three strains without S9, with the oxidative mutagen-sensitive strains TA102 and TA104 showing higher responses than TA100. While dark roast drip coffees showed higher mutagenicity per gram, instant coffee was more mutagenic per cup (20,000 revertants vs. 2500 revertants) when actual serving sizes were considered. Decaffeinated and caffeinated instant coffees showed similar mutagenic responses; this indicated that caffeine is not the primary mutagen. With S9, mutagenicity was significantly reduced or eliminated in TA100 for most coffees, but persisted in TA102 and TA104 for most samples; decaffeinated coffees even showed enhanced mutagenicity in TA104 with S9. Both glyoxal and methylglyoxal, identified as components of the coffees, were also found to be mutagenic. Methylglyoxal was estimated to account for 33–55% of the observed mutagenic activity, with an inverse relationship noted between its concentration and mutagenic response in TA102/TA104, suggesting a possible synergistic effect with H_2_O_2_. These results indicate direct-acting mutagens in all coffee types, with carbonyl compounds contributing to, but not fully explaining, the mutagenicity.

Kato et al. [47] investigated the DNA-breaking and mutagenic activity of purified instant coffee. DNA-breaking activity was assessed using commercial plasmid pBR 322 DNA, and mutagenicity was tested using the Ames assay. For purification, 250 mg of instant coffee was dissolved in 1.0 mL of water, centrifuged, and the 0.5 mL supernatant was applied to a Sephadex G-10 column, collecting 0.5 mL fractions. A 0.1 mL aliquot of each fraction was then tested for mutagenicity. Purified instant coffee was shown to cleave DNA, causing single-strand breaks. While all fractions were non-mutagenic to TA100, fractions 2–4 exhibited significant mutagenicity to TA98 without metabolic activation, with an estimated 4000 His^+^ revertant colonies per gram of instant coffee powder. Brewed and instant coffee exhibited strong chemiluminescence, attributed to singlet oxygen and excited carbonyls likely generated by the Maillard reaction of sugars and amino acids. The authors concluded that DNA-breaking activity was not attributed to chemiluminescent materials or active oxygen radicals, as demonstrated by the marginal effects of active oxygen radical scavengers on the active gel fraction’s DNA-breaking activity. This activity was, however, inhibited by high concentrations of inorganic salts, likely due to DNA double-strand stabilization. The study suggested that the DNA-breaking activity and mutagenicity, observed after purification by gel filtration, were masked in whole coffee by other components, likely inorganic salts.

In accordance with previous findings, instant coffee exhibited direct-acting mutagenicity in TA100, and this effect was also observed in TA102 [48]. Adding rat liver homogenate (10% S9 mix) completely eliminated this activity, with even greater reduction when reduced glutathione was also added, even with inactivated S9. The substance responsible for this inactivation was heat-sensitive and found in the S9′s cytosol. Mutagenicity also decreased spontaneously when the extracts were incubated at 0–50 °C (with ~50% reduction at 50 °C after 6 h), with faster reduction at higher temperatures; this reduction was prevented in the absence of oxygen, suggesting oxidative inactivation. Low pH (1–3) did not affect the mutagen’s stability. Comparing coffee’s effects to methylglyoxal, a potential coffee mutagen, showed that methylglyoxal was also strongly mutagenic to both strains, partially inactivated by 10% S9, and completely inactivated by glyoxalase I/II and reduced glutathione, which only partially reduced coffee’s mutagenicity (by 80%), suggesting other mutagens are present in coffee.

The mutagenic effects of coffee, with a focus on the role of oxidative stress and particularly H_2_O_2_, were investigated by Ariza et al. [43]. The study compared the mutagenicity of ground and instant coffee using the *Salmonella* Ara test with strain BA13 (araD531, hisG46, ΔuvrB, pKM101) without metabolic activation (S9). Instant coffee exhibited significantly higher mutagenicity than ground coffee. Agitation of bacterial cultures during the Ara test increased mutagenicity for both coffee types, attributed to reduced catalase activity in exponentially growing cells, enhancing sensitivity to coffee-derived ROS, particularly H_2_O_2_. The higher mutagenicity of instant coffee is likely due to its greater H_2_O_2_ concentration. Addition of catalase significantly reduced the mutagenicity of both coffee types, confirming the role of oxidative stress. Reactive carbonyl compounds like methylglyoxal may synergistically amplify the mutagenic effects of H_2_O_2_, especially in instant coffee. The study concluded that instant coffee is more mutagenic than ground coffee in the Ara test, primarily due to H_2_O_2_ and promoted by agitation, emphasizing the role of oxidative stress and preparation methods in determining coffee’s mutagenic potential.

Ariza and Pueyo [49] reported similar findings, demonstrating that instant coffee is mutagenic in the *Salmonella* L-arabinose resistance test without S9 activation, indicative of directly acting mutagens. S9 effectively neutralized this mutagenicity, reducing it to 3% of its original level. Catalase was identified as the key enzyme within S9 responsible for this neutralization.

In a similar study, Duarte et al. [50] investigated the genotoxicity of instant coffee using the Ames assay. Initial testing was performed at concentrations of 0–40 mg/plate, followed by more specific testing in strain TA100 at pH 6.3, 7.4, and 8.1 using 0–15 mg/plate. Instant coffee exhibited direct genotoxic activity mediated by ROS, primarily H_2_O_2_, and was observed in all strains tested. S9 metabolic activation, particularly catalase, significantly reduced or abolished genotoxicity: completely in TA104 and YG1024, reduced in TA98 and TA100, and slightly reduced in TA102. Without S9, significant genotoxicity was observed in TA98, TA100, and TA102, with dose-response effects up to 15 mg/plate. With S9, the genotoxicity threshold increased to 40 mg/plate. Genotoxicity increased with pH, peaking above neutrality (pH 8). The study attributed the observed effects to H_2_O_2_ production via auto-oxidation of phenolic compounds, which is enhanced at alkaline pH due to deprotonation and reaction with oxygen.

In a complementary study, Stadler et al. [19] used the Ames plate assay with *S. typhimurium* strains TA100 and TA102 to investigate the genotoxicity of instant coffee, with a focus on the role of catalase. At concentrations up to 10 mg/plate, no significant increase in revertants was observed. However, at concentrations above 10 mg/plate, increased toxicity and mutagenic effects (induction of revertants) were observed. The addition of catalase, which degrades H_2_O_2_, abolished both the toxic and mutagenic effects, confirming the crucial role of H_2_O_2_ in coffee’s mutagenic and bactericidal activity.

Fujita et al. [51] reported that instant coffee exhibits a linear dose-response for mutagenicity in *S. typhimurium* TA100 up to cytotoxic levels (≥20 mg/plate). Methylglyoxal’s specific mutagenic activity is 120 revertants/µg in *S. typhimurium* TA100, but its effect is non-linear at lower doses. The study concludes that while methylglyoxal exhibits mutagenic activity on its own, its contribution to instant coffee’s overall mutagenicity is minor, suggesting the presence of other significant mutagenic compounds in coffee that warrant further investigation. This was confirmed through suppression experiments using cysteine (specific to methylglyoxal) and catalase (specific to other coffee components). Cysteine suppresses the mutagenicity of added methylglyoxal to coffee but not coffee’s inherent mutagenicity. Catalase suppresses most of coffee’s mutagenicity but not methylglyoxal’s, whether added to coffee or alone.

Nakasato et al. [44] also showed that while coffee contains the mutagen methylglyoxal, this compound accounts for less than 3% of coffee’s overall mutagenicity. The diphtheria toxin resistance assay revealed the mutagenic activity of instant coffee in cultured Chinese hamster lung (CHL) cells. Coffee induces mutations in these cells, resulting in a dose-dependent increase in DT^r^ cells. Methylglyoxal exhibits concentration-dependent mutagenicity in CHL cells. However, the study confirmed that this compound contributes minimally to coffee’s overall mutagenicity. Importantly, caffeine, a known component of coffee, does not demonstrate mutagenic activity in CHL cells at tested concentrations, indicating that other compounds present in coffee are responsible for the observed cellular effects. The observation that sodium bisulfite, a carbonyl scavenger, reduces coffee’s mutagenicity in a dose-dependent manner suggests the involvement of reactive carbonyl compounds in coffee’s genotoxic effects.

Hossain et al. [61] reported that dietary concentrations of coffee (regular and decaffeinated) caused significant DNA strand breaks using p53R cells, a cellular assay sensitive to such breaks. The study confirms that p53 is activated in response to DNA strand breaks. At approximately 5 µg/mL, etoposide (the positive control) induced a similar DNA damage response as a 1:20 dilution of coffee, suggesting comparable levels of DNA damage. Pyrogallol, 3-methoxycatechol, gallic acid, and 1,2,4-benzenetriol also exhibited strong DNA-breaking activity, highlighting the role of hydroxyl group orientation on the benzene ring for this activity. Pyrogallol triggered the phosphorylation of H2AX (γ-H2AX) in a dose- and time-dependent manner, which is a marker of strand breaks.

Duarte et al. [41] investigated the genotoxic potential of nitrosated instant coffee using the Ames test, considering that coffee contains phenolic and nitrogenous compounds capable of reacting with nitrite under acidic conditions, simulating the gastric environment. Nitrosated instant coffee exhibited concentration-dependent genotoxicity, while non-nitrosated coffee showed no genotoxicity. While nitrosated chlorogenic acid, catechol, and caffeic acid were also genotoxic, their contribution to the overall genotoxicity of nitrosated coffee was considered minor due to their relatively low concentrations in coffee. These results showed that nitrosation of instant coffee generates genotoxic compounds, likely from both specific phenolic compounds and other unidentified nitrosatable molecules. However, phenolic nitrite scavengers present in coffee may offer some natural protection, and the in vivo relevance of these genotoxic effects remains uncertain due to limited nitrite availability in the human body.

#### 3.3.3. Coffee By-Products

Fernandes et al. [64] investigated the mutagenic and genotoxic potential of leached (LE) and solubilized (SE) extracts from spent coffee grounds to simulate environmental disposal. Mutagenicity was assessed using the Ames test, and genotoxicity with the MN assay in HepG2 cells. In the Ames test, both LE and SE were mutagenic to TA98 with and without S9. For TA100, LE was not mutagenic but was toxic with S9, while SE was mutagenic only without S9. Both LE and SE induced micronuclei formation in HepG2 cells in a dose- and time-dependent manner (3 and 24 h), with LE showing higher micronuclei formation than SE at 24 h and SE showing a 10-fold increase at 6.25% after 3 h. LE showed greater genotoxic effects, particularly with increased treatment time, highlighting the potential formation of genotoxic compounds during the leaching process. The study found that caffeine and several transition metals were present in both extracts, potentially contributing to the observed effects. The interaction of metals, caffeine, and polycyclic aromatic hydrocarbons could synergistically enhance mutagenicity and toxicity. Despite their hazardous effects, some toxicity could be mitigated by metabolic detoxification processes. The results showed that improper disposal of coffee waste poses mutagenic and genotoxic potential, highlighting the need for effective waste management strategies.

#### 3.3.4. Coffee Bioactive Compounds

The clastogenic potential of mangiferin, a compound found in *C. arabica* leaves, was evaluated using an in vitro mammalian chromosomal aberration assay, which confirmed its clastogenic effect. Evidence suggests that polyphenols can induce chromosomal aberrations in in vitro chromosomal aberration test at low concentrations. Mangiferin may have a similar effect, requiring further study for confirmation [82].

Wu et al. [67] investigated the mutagenic potential of heated trigonelline and its interactions with amino acids, components naturally present in green coffee beans, using bacterial mutation assays (*S. typhimurium* strains TA98, YG1024, and YG1029). Trigonelline, either alone or combined with single amino acids or mixtures of amino acids and glucose, was heated at 250 °C for 20 min to simulate roasting. Trigonelline alone exhibited the highest mutagenic activity among singly heated compounds. In experiments with single amino acids, all combinations with trigonelline were mutagenic except those with cystine and tryptophan. Experiments with amino acid mixtures (with and without S9) showed mutagenicity in all cases except for YG1029 with S9. Mutagenicity was higher without S9, indicating the predominance of direct-acting mutagens, while mutagenicity with metabolic activation suggested the formation of amines as potential mutagenic compounds.

Müller et al. [88] investigated the mechanism by which caffeine, at a concentration that did not induce DNA damage, enhances radiation risk in mammalian cells during the two-cell stage of gestation, using the comet assay. The study found that caffeine increases the radiation risk by a factor of 1.5 to 2, primarily by inhibiting DNA repair. At 2 mM, caffeine significantly inhibited the restitution of radiation-damaged DNA, leaving over 40% of the initial damage unrepaired after 2 h, compared to near-complete repair in caffeine-free conditions. The study concluded that physiologically relevant concentrations of caffeine (2 mM) effectively inhibit DNA repair, explaining its enhancement of radiation-induced DNA damage, particularly relevant in early embryonic development where DNA repair is crucial.

Kim et al. [84] showed that carbonyl compounds (diacetyl, methyl glyoxal, glyoxal, HMF, acetol and furfural) induced base-substitution mutations in *S. typhimurium* TA100 without metabolic activation. Methyl glyoxal exhibited the strongest mutagenic activity among those tested.

The clastogenic activity of various carbohydrate pyrolysates was tested in Chinese hamster V79 cells [85]. Furan derivatives (furfural and HMF) and glyoxal derivatives (glyoxal and methylglyoxal) caused significant chromosome aberrations and reduced mitotic activity. Furfural and methylglyoxal exhibited stronger clastogenic activity than HMF and glyoxal, respectively.

Janzowski et al. [93] investigated the genotoxicity of HMF, a compound formed during coffee roasting, using umu assay (SOS response in *S. typhimurium* TA1535/pSK1002) and the comet assay for mammalian DNA damage. HMF directly induced a DNA damage response in *S. typhimurium* without metabolic activation (16 mM HMF: 185 relative umu units, *p* < 0.001), unaffected by the addition of rat liver microsomes. However, weak mutagenicity was observed at the hprt-locus in V79 cells at 120 mM HMF (16 mutants/10^6^ cells vs. 3 mutants/10^6^ cells in the control) that already are moderately cytotoxic. While HMF exhibits mutagenic, and genotoxic effects at high concentrations, these levels are significantly higher than those typically found in food. In addition, compared to (E)-2-hexenal (HEX), a natural flavor compound, HMF was significantly less toxic and genotoxic, as HEX exerted effects at concentrations ~1000 times lower.

Surh et al. [100,101] found that the mutagenicity of HMF results from its biotransformation into 5-sulfoxymethylfurfural (SMF) by sulfotransferases (SULTs). SMF is a reactive allyl ester that is mutagenic in bacteria and mammalian cells without requiring an activating system. SMF’s mutagenicity is linked to the formation of a highly electrophilic allyl carbocation, which reacts with cellular macromolecules. SMF’s genotoxic effects were mitigated by glutathione (GSH) and glutathione *S*-transferases (GST), which inactivate SMF [99].

However, Durling et al. [95] showed that HMF-induced DNA damage is independent of SULT1A1 expression. In the in vitro comet assay, HMF induced significant DNA damage in all tested cell lines after a 3-h exposure to 100 mM. While V79-hP-PST cells (high SULT1A1 expression) were more sensitive than parental V79 cells, no correlation was found between DNA damage and SULT1A1 activity.

Severin et al. [94] assessed the genotoxic potential of HMF in HepG2 cells using the comet assay. Their findings revealed that HMF induced weak DNA damage at concentrations ranging from 7.87 to 25 mM.

An analysis of ground roasted coffee bean extract identified HMF, acetol, glyoxal, methylglyoxal, and diacetyl as major carbonyl compounds. Of these, methylglyoxal exhibited significant mutagenic activity in *S. typhimurium* TA100 without S9 mix (approximately 100,000 revertants/mg), accounting for more than 50% of coffee’s total mutagenic activity [87].

Tucker et al. [86] evaluated the genotoxicity of 1,2-dicarbonyl compounds (e.g., glyoxal, methylglyoxal, kethoxal) found in coffee using CHO cells and human lymphocytes. Glyoxal, methylglyoxal, and kethoxal induced sister chromatid exchanges (SCEs), indicating DNA damage. H_2_O_2_ also increased SCEs, highlighting oxidative stress as a key factor. Bisulfite reduced the genotoxic effects, suggesting that reactive carbonyl compounds play a role in food-related genotoxicity.

Nagao et al. [83] showed that coffee’s mutagenicity is not solely due to methylglyoxal, as it differs from the effects of pure methylglyoxal. Coffee contains mutagens that induce 5 × 10^4^–10^5^ revertants in *S. typhimurium* TA100 without S9 mix. Coffee’s mutagenicity is suppressed by catalase and is associated with the presence and in situ generation of H_2_O_2_ (130 µM in 15 mg/mL instant coffee immediately after dissolution, increasing over time). H_2_O_2_ enhances methylglyoxal’s mutagenicity up to 30-fold. Glyoxal, propylglyoxal, and ethylglyoxal, also present in coffee, exhibited lower mutagenic activity in TA100 than methylglyoxal and are found in smaller quantities.

The mutagenicity of chemicals commonly found in coffee solutions—H_2_O_2_, methylglyoxal, glyoxal, caffeic acid, and chlorogenic acid—was investigated by Ariza et al. [43] using the Ara test, which is more sensitive to oxidative mutagens like H_2_O_2_ than the Ames test. H_2_O_2_ was identified as a strong mutagen, contributing 40–60% of coffee’s mutagenicity; this mutagenicity was largely abolished (>95%) by catalase. H_2_O_2_ showed synergistic mutagenic effects when combined with methylglyoxal or glyoxal, especially at higher doses. Methylglyoxal, glyoxal, and caffeic acid were classified as intermediate mutagens, while chlorogenic acid was a weak mutagen. The minimal mutagenic doses of individual compounds (excluding H_2_O_2_) were much higher than their concentrations in coffee, suggesting minor direct contributions to coffee mutagenicity, except for H_2_O_2_, whose concentration aligned with its mutagenic impact.

In a separate study, synergistic mutagenic effects of H_2_O_2_ with methylglyoxal were observed using the Ames test; H_2_O_2_ alone, however, showed weak mutagenicity [89]. This H_2_O_2_-generating system is a product of the roasting process, as green coffee beans exhibit very low H_2_O_2_ generation capacity.

However, Aeschbacher et al. [90] suggested that coffee’s mutagenicity is likely due to the combined effects of several compounds, such as aliphatic dicarbonyls, rather than a synergistic interaction between methylglyoxal and H_2_O_2_. They examined the mutagenic potential of coffee aroma constituents belonging to the classes of dicarbonyls, sulphur-containing compounds, furfuryls, and *N*-heterocyclics using Ames test. Of approximately 40 compounds tested, only aliphatic dicarbonyls (specifically methylglyoxal, glyoxal, and diacetyl) showed reproducible and significant mutagenic effects, particularly in TA100 and, to a lesser extent, TA102, both with and without metabolic activation. Other compounds show minimal or no mutagenicity. Quinoxaline was the only *N*-heterocyclic to show a weak but significant effect in TA98 (+S9). Levels of H_2_O_2_ and methylglyoxal, along with mutagenic activity in strains TA100 and TA102, decreased over time.

Ruiz-Laguna and Pueyo [52] compared the mutagenic effects of H_2_O_2_ and coffee (instant, regular roast), hypothesizing that H_2_O_2_ contributes significantly to coffee’s mutagenic effects using the Ara test and *lacI* mutant analysis in *E. coli*. The study used catalase-proficient (UC838) and catalase-deficient (katG katE mutant, UC1221) strains, as well as strains with a mutagenesis-enhancing plasmid (pRW144, UC1218 and UC1218 katG katE). H_2_O_2_ and instant coffee induced similar mutagenic profiles, primarily G:C→T:A transversions, in catalase-deficient strains via 8-oxo-guanine lesions. The similar distribution of mutational events induced by both H_2_O_2_ and coffee suggests that H_2_O_2_ is a major contributor to coffee-induced mutagenesis. This mutagenicity was significantly amplified by the pRW144 plasmid. The Ara test also showed a dose-response relationship for both coffee and H_2_O_2_, with higher concentrations leading to more L-arabinose-resistant mutants, especially in catalase-deficient strains. Although the overall frequency of G:C→T:A transversions was similar between H_2_O_2_ and coffee, the different locations of these mutations within the *lacI* gene suggested contributions from other coffee components.

Yoshie et al. [73] reported that incubation of plasmid DNA (pBR322) with an NO-releasing compound (diethylamine NONOate containing a [N(O)NO]^−^ moiety) and a polyhydroxyaromatic compound (catechol, pyrogallol, or chlorogenic acid) resulted in a synergistic increase in single-strand DNA breaks, whereas each compound alone caused minimal damage. This synergistic interaction may be particularly relevant in pathological conditions where both are present in vivo.

Fung et al. [68] evaluated the mutagenic activity of caffeic acid and chlorogenic acid using the L5178Y TK⁺/^−^ mouse lymphoma assay. Caffeic acid showed a five-fold increase in mutant frequency compared to the control but only without metabolic activation, while chlorogenic acid showed an 8-fold increase but only with metabolic activation. These results were consistent with previous findings of caffeic and chlorogenic acid genotoxicity in other mammalian systems such as gene conversion in *Saccharomyces cerevisiae* and chromosomal aberrations in CHO cells [69,74].

Stich et al. [69] investigated the genotoxicity of chlorogenic acid using three test systems: the *S. typhimurium* mutagenicity assay, the *S. cerevisiae* gene conversion assay, and the chromosome aberration test in CHO cells. Chlorogenic acid showed mutagenic activity in *S. typhimurium* strains TA98 and TA100 only in the presence of Mn^2+^ (10^−4^ M), suggesting a role for transition metals. In the yeast gene conversion assay, chlorogenic acid demonstrated convertogenic activity, which was enhanced by Mn^2+^ but suppressed by the S9 metabolic activation system. The compound also exhibited direct clastogenic activity in CHO cells, causing chromatid breaks and exchanges, with Mn^2+^ significantly amplifying these effects, while the S9 mix reduced or abolished the activity. Components of chlorogenic acid, such as caffeic acid, showed similar genotoxic effects, while quinic acid was less potent. These findings underscore the importance of employing multiple genotoxicity assays to capture various types of genetic damage and highlight the role of transition metals like Mn^2+^ in modulating the genotoxicity of compounds.

Micromolar concentrations of Cu^2+^ induced DNA strand breaks in the presence of various phenolic compounds including caffeic acid and chlorogenic acid. Further investigation indicated that singlet oxygen or a singlet oxygen-like entity, possibly a copper-peroxide complex, rather than free hydroxyl radicals, likely plays a key role in the DNA damage. These findings suggest that macromolecule-associated copper and reactive oxygen generation are important factors in the mechanism of DNA damage induced by phenolic compounds in target cells [70].

Using isolated λ DNA systems, Yamada et al. [75] investigated DNA breakage induced by phenyl compounds found in food, including chlorogenic acid. Chlorogenic acid was observed to induce DNA double-strand breaks at a concentration of 250 µM.

The genotoxicity of several plant phenolics including chlorogenic acid across a pH range of 5 to 10 using *S. cerevisiae* strain D7 was also investigated by Rosin et al. [76]. The key finding was that these substances induced mitotic conversion only at alkaline pH levels, while showing no such effect at acidic pH. This pH-dependent genotoxicity is likely attributable to the accelerated autoxidation of phenolics under alkaline conditions, leading to the formation of H_2_O_2_ and free radicals.

Chlorogenic acid was also reported to induce genotoxicity in Chinese hamster V79 cells at 0.07 mg/mL by Pereira et al. [77].

Whitehead et al. [78] reported similar results, showing that chlorogenic acid (0.25 mg/mL) and caffeic acid (0.28 mg/mL) induced chromosomal aberrations (clastogenic effects) in Chinese hamster ovary (CHO) cells. The addition of intestinal epithelial cells or S9 metabolic activation mixture abolished the chromosome-damaging activity of these compounds.

Burgos-Morón et al. [20] investigated the effects of chlorogenic acid (CGA) on DNA damage in K562 cells using the comet assay and γ-H2AX focus assay. The study found that CGA induces DNA damage in both normal and cancer cells in a dose-dependent manner, as shown by both assays. CGA also induced significant levels of topoisomerase I-DNA and topoisomerase II-DNA complexes, as demonstrated by the TARDIS assay, suggesting interference with topoisomerase activity. Pretreatment with catalase, which degrades H_2_O_2_, reduced both topoisomerase–DNA complex formation and CGA’s cytotoxic activity, highlighting the central role of H_2_O_2_ in CGA’s effects.

Hiramoto et al. [66] identified hydroxyhydroquinone (HHQ) as a DNA-damaging component in instant coffee, capable of inducing single-strand DNA breaks in supercoiled pBR322 DNA at 0.1 mM. The damage was attributed to ROS, primarily H_2_O_2_ and hydroxyl radicals. Catalase, hydroxyl radical scavengers (mannitol, potassium iodide, sodium azide, ethanol), spin-trapping agents (DMPO, PBN), and thiol compounds (2-mercaptoethanol, cysteine) significantly reduced HHQ-induced DNA damage, confirming the role of ROS. Unlike other coffee polyphenols (e.g., hydroquinone and pyrogallol) and Maillard reaction products, HHQ exhibits strong genotoxic activity, which is enhanced by copper ions.

Furan forms during roasting of green coffee beans. Sulfotransferases play a critical role in the bioactivation of hydroxymethyl-substituted furan, as reported by Glatt et al. [96]. The study examined the mutagenicity of HMF, furfuryl alcohol (FFA), and their metabolites in *S. typhimurium* TA100 and strains expressing human or rodent sulphotransferases (SULTs). HMF, 2,5-(bishydroxymethyl)furan, and FFA were non-mutagenic in standard TA100 but showed strong mutagenicity in strains expressing human SULT1C2, an enzyme present in tissues like the ovary, kidney, and fetal tissues. 3-Hydroxymethylfuran exhibited weak mutagenicity across all strains regardless of SULT expression, while 5-hydroxymethylfuroic acid was not mutagenic. Furfuryl sulphate showed bacteriotoxicity and only marginal mutagenicity due to its short half-life (20 s at 37 °C in water) and limited cellular penetration. However, furfuryl sulphate was mutagenic when bioactivated intracellularly by SULTs from FFA. These findings also emphasize the need for tissue-specific evaluations of these compounds.

Coffee Maillard reaction products (MRPs) were tested for their effect on Fe^2+^-induced DNA strand breakage under various pH conditions by Wijewickreme and Kitts [98]. Double-stranded supercoiled DNA from bacteriophage PM2 served as a biological model to measure the extent of oxidative DNA strand breaks. Supercoiled DNA was converted into nicked circular and linear forms, indicating strand scissions. Results showed that MRPs and Fe^2+^ alone induced DNA strand breaks in a dose-dependent manner at all tested pH conditions, with greater damage occurring at lower pH values. The findings indicate that coffee MRPs can act as both antioxidants and pro-oxidants depending on their concentration. At low doses, MRPs chelate Fe^2+^, reducing its ability to generate ROS and mitigating DNA damage. However, at higher concentrations, MRPs lost their protective ability and instead promoted DNA breakage, possibly by influencing the redox state of Fe^2+^.

### 3.4. Studies Showing Protective and Antimutagenic Effects

The antigenotoxic effects of caffeinated and decaffeinated instant coffee, as well as boiled coffee (coarse ground roasted coffee was boiled in water and hot-filtered), against MNNG-induced genotoxicity was evaluated in mouse lymphoma L5178Y cells using multiple assays: the comet assay, Tk locus mutation assay, and cytokinesis-block micronucleus assay. The preparation of boiled coffee involved boiling coarse ground roasted coffee in water, followed by hot filtration. All types of coffee demonstrated significant antigenotoxic effects, with no notable differences between caffeinated and decaffeinated instant coffee. Caffeinated instant coffee exhibited dose-dependent inhibition of genotoxicity in the micronucleus assay, and filtration did not affect its protective activity. Boiled coffee showed antigenotoxic effects comparable to instant coffee. These findings confirm that both instant and boiled coffee, regardless of caffeine content, possess strong antigenotoxic properties, reducing MNNG-induced DNA and chromosomal damage [25].

Protective effects of instant coffee against genotoxic agents in L5178Y mouse lymphoma cells using micronucleus and comet assays was demonstrated in another study by Abraham et al. [55]. Coffee significantly reduced chromosomal damage caused by MNNG, mitomycin C (MMC), methyl methanesulfonate (MMS), and gamma radiation, particularly at higher radiation doses (1 and 2 Gy). Comet assay confirmed a reduction in MMS-induced DNA damage when cells were co-treated with coffee. However, coffee had only a minor effect on apoptosis, suggesting that its protective action is not solely due to increased cell death sensitivity.

Additionally, the protective effects of instant coffee against oxidative damage were assessed using the Ames plate assay with *S. typhimurium* strains TA100 or TA102. Low concentrations of instant coffee (2.5–20 mg/plate) reduced the number of revertants induced by t-butylhydroperoxide (t-BOOH) by 40–90%, demonstrating an antioxidant or protective effect against oxidative damage [19].

Bichler et al. [59] investigated the DNA-protective effects of coffee (metal-filtered coffee with French press method) consumption in peripheral human lymphocytes by comet assay and confirmed its antioxidant and genoprotective properties. Coffee did not induce DNA damage in lymphocytes. Instead, coffee significantly reduced oxidative DNA damage, lowering oxidized purines (64%), oxidized pyrimidines (48%), and H_2_O_2_-induced DNA damage (17%). It also protected against Trp-P-2-induced DNA damage, likely by inducing detoxification enzymes (GST, UDP-glucuronosyltransferase). Additionally, it increased SOD enzyme activity by 38%. Diterpenoids cafestol and kahweol (C + K) exhibited only moderate protection (27–38%) compared to whole coffee extracts, suggesting that other coffee components contribute to DNA protection.

Glei et al. [35] confirmed that H_2_O_2_ induces DNA damage in HT29 and HepG2 cells using the alkaline comet assay, but chlorogenic acid, and green coffee, provided significant protection. Chlorogenic acid reduced DNA damage at ≥100 µM (HT29) and ≥10 µM (HepG2) within 30 min, while long-term exposure (24 h) required higher doses (500 µM). Green coffee only showed protective effects after 24 h, significantly reduced DNA strand breaks at ≥0.006 g/L green coffee. The neutral comet assay revealed that only chlorogenic acid reduced double-strand breaks, with 1 µM chlorogenic acid lowering damage by 47% (30 min) and 58% (24 h), confirming its strong chemoprotective potential against oxidative stress.

Using the comet assay, Bravo et al. [65] reported the protective effects of spent coffee grounds (SCG) extracts (Arabica filter and Canephora espresso) against oxidative stress in HeLa cells. Both extracts significantly reduced H_2_O_2_-induced DNA strand breaks (29–73%) and ROS levels. Pretreatment with Canephora extract further reduced oxidative DNA damage caused by photosensitizers after 24 h. Despite containing fewer caffeoylquinic acids and melanoidins, Canephora extract was more effective in minimizing DNA damage, suggesting additional protective compounds are involved. These findings highlight SCG extracts’ potential as natural agents to combat oxidative stress.

Monente et al. [62] reported similar results, testing the antimutagenic effects of spent coffee grounds (SCG) and coffee brews (Arabica filter and Canephora espresso) against the direct mutagen 4-Nitro-o-phenylenediamine (NPD) and the indirect mutagen 2-aminofluorene (2-AF) using the Ames test with and without S9 metabolic activation. Both SCG extracts and brews reduced mutagenicity against both mutagens, with SCG extracts being more effective due to higher levels of bioactive compounds. Against NPD (without S9), Canephora SCG exhibited the greatest reduction in mutagenicity, decreasing it by up to 35%, likely through free radical scavenging by phenolic compounds and caffeine. Against 2-AF (with S9), both SCG and brews provided strong protection (up to 92%) by interfering with 2-AF metabolic activation through scavenging free radicals, inhibiting enzymes like cytochrome P450, and competitive inhibition by caffeine. Overall, SCG extracts and brews showed two protective mechanisms: direct action on reactive species and indirect action by disrupting mutagen activation, with SCG extracts being the most effective.

Iriondo-DeHond et al. [63] evaluated the protective effects of coffee silverskin extract and chlorogenic acid against benzo[*a*]pyrene (BaP)-induced DNA damage in HepG2 cells using the alkaline and enzyme-modified comet assays. Both silverskin extract and chlorogenic acid significantly reduced BaP-induced DNA strand breaks at all concentrations tested, with maximum reductions of 26% and 29%, respectively, at 100 μg/mL. They also significantly reduced oxidative DNA damage, including FPG-sensitive sites (oxidized purines) and Endonuclease III-sensitive sites (oxidized pyrimidines). The maximum reductions observed were 69% (silverskin extract) and 76% (chlorogenic acid) for FPG-sensitive sites and 70% (silverskin extract) and 77% (chlorogenic acid) for Endo III-sensitive sites. These findings suggest that both silverskin extract and chlorogenic acid, due to their antioxidant properties, effectively protect against BaP-induced DNA damage and oxidative lesions, highlighting their potential as natural chemoprotective agents against chemical carcinogens.

Coffee-specific diterpenes cafestol and kahweol (C + K) are anticarcinogenic likely due to their ability to modulate carcinogen detoxification enzymes, reducing DNA damage from reactive agents. C + K significantly reduced the formation of AFB1-DNA adducts in human liver epithelial cells (THLE) cells expressing aflatoxin B_1_-metabolizing cytochrome P450s. Protection was linked to the induction of GST (glutathione *S*-transferase) and AKR (aldo-keto reductase) enzymes, which aids in detoxifying AFB1-8,9 epoxide (AFBO) [92]. To investigate whether coffee directly inhibits the activation of AFB1, human primary hepatocytes were treated simultaneously with varying concentrations of coffee (caffeinated and decaffeinated) and 10 nM tritiated AFB1. Coffee significantly reduced AFB1-DNA adduct formation in vitro in a dose-dependent manner. The results demonstrated that coffee inhibited CYP1A1/2, key enzymes in the cytochrome P450 family, thereby reducing the bioactivation of carcinogens. In addition, coffee activated Nrf2-ARE signaling, enhancing cellular defense against electrophilic and oxidative stress, further reinforcing its potential role in protecting against chemical-induced damage [58].

Majer et al. [91] explored the genotoxic and chemoprotective effects of coffee diterpenoids, cafestol palmitate, and a mixture of cafestol and kahweol (C + K), against the dietary carcinogens *N*-nitrosodimethylamine (NDMA) and 2-amino-1-methyl-6-phenylimidazo[4,5-*b*]pyridine (PhIP) in HepG2 cells using micronucleus assays. Neither compound exhibited direct genotoxicity. Both provided significant protection against PhIP-induced genotoxicity, with maximum inhibition observed at 0.9 µg/mL (C + K) and 1.7 µg/mL (cafestol palmitate), through reducing PhIP activation by inhibition of sulfotransferase, enhancing detoxification by induction of UDP-glucuronosyltransferase, and increased glutathione *S*-transferase activity. C + K also reduced NDMA-induced genotoxicity by 50% at low doses (0.3 µg/mL), likely by inducing DNA repair enzymes. The findings suggest that coffee diterpenoids are not genotoxic and may act as natural chemoprotective agents against dietary carcinogens such as PhIP and NDMA through enzyme modulation and DNA repair.

Using the comet assay, Edenharder et al. [57] demonstrated that coffee reduced PhIP genotoxicity in Chinese hamster lung fibroblasts (V79 rCYP1A2-rSULT1C1), suggesting that various antigenotoxic compounds in coffee can penetrate V79 cell membranes and act intracellularly.

Molund et al. [16] investigated the potential of the water-insoluble fraction (WIF) of coffee to inhibit mutagenesis induced by known mutagens using the Ames test. The mutagens used were AFB1, MNNG, and BaP. WIF dramatically inhibited AFB1-induced mutagenesis in the presence of S9, with significant inhibition observed at a WIF concentration of 3 mg/mL. In the absence of AFB1, WIF itself did not induce significant mutagenesis. Inhibition of MNNG-induced mutagenesis increased significantly with the addition of WIF at 5 mg/mL, with further increases up to 25 mg/mL. A gradual inhibition of BaP-induced mutagenesis was observed with increasing WIF concentrations up to 30 mg/mL. The proposed mechanisms for this inhibition are: (1) Adsorption of carcinogens, reducing their bioavailability. (2) Adsorption of cofactors/enzymes within the monooxygenase system (present in the S9 mix), thus reducing metabolic activation of promutagens like AFB1 and BaP. (3) Quenching of free radicals generated by the mutagens or during their metabolic activation.

## 4. Discussion

Different types of coffee products and their processing methods significantly influence their genotoxic profiles. For example, coffee fruit, green coffee and brewed coffee often show a lower genotoxic potential compared to instant coffee or coffee processed at high temperatures [15,32,33,39]. While coffee may exhibit mutagenicity in vitro, metabolic processes likely neutralize any potential risk in living organisms [90]. A long-term rat study found no evidence of carcinogenicity from instant coffee, suggesting that in vitro mutagenicity does not always mean in vivo cancer risk [48].

The increasing use of coffee by-products in functional foods and sustainable materials highlights the need for comprehensive safety assessments. Spent coffee grounds and silverskin, though rich in bioactive compounds, have shown both beneficial and harmful biological effects depending on their concentration and mode of application and further safety assessments are required to evaluate their long-term effects [7,64,102].

There are conflicting findings regarding coffee’s genotoxicity. In addition, human epidemiological data indicate an inverse relationship between coffee intake and certain cancers [27], further complicating the interpretation of in vitro genotoxicity results. However, discrepancies may arise due to differences in experimental conditions. For example, some studies used several genotoxicity tests with different end points, or examine a wide range of concentration of samples that provide a more comprehensive and reliable assessment of genotoxicity.

The genotoxicity of caffeic acid and chlorogenic acid varies depending on experimental conditions. While they can act as antioxidants, they may also exhibit pro-oxidant behavior particularly in the presence of transition metals, such as iron, leading to ROS production and oxidative stress that damages DNA. Consequently, it is difficult to predict whether they will primarily act as pro-oxidants or antioxidants in a cell system [35]. In a study of Hernandes et al. [71], HL-60 and Jurkat cells were cultured in RPMI 1640 medium, which has low iron levels, potentially minimizing oxidative stress. As a result, no genotoxic effects were observed for caffeic and chlorogenic acid. However, previous research [103] found that in media, which contain iron, these phenolic acids underwent oxidation, increasing H_2_O_2_ production and promoting DNA damage. This suggests that iron catalyzes their auto-oxidation, generating ROS. In studies of Burgos-Morón et al. [20], Stich et al. [69], and Pereira et al. [77], where chlorogenic acid exhibited genotoxic activity, cell lines were cultured in media, which contain ferric nitrate. Thus, contradictory findings on the effects of phenolic acids may be due to differences in culture medium composition, particularly iron availability, which influences their antioxidant or pro-oxidant behavior.

Chlorogenic acid (CQA) shows inconsistent mutagenic effects across different test systems, depending on several factors including whether the DNA is isolated or within a cell, the presence of repair mechanisms, and external conditions like pH and metal ions. In acellular systems (isolated DNA and supercoiled ΦX174 RF I DNA), it induces DNA double-strand breaks, especially in the presence of NO-releasing agents or Cu^2+^ [70,73,75]. However, in *S. typhimurium* assays, it is non-mutagenic, likely due to cellular repair mechanisms [41,68,69,79,80] and exhibited genotoxic activity only in the presence of Mn^2+^ and under nitrosation [41,69]. In *S. cerevisiae* (strain D7), CQA induces mitotic gene conversion at high concentrations [69] and at low concentrations only under alkaline pH [76].

The choice of bacterial strains affects mutagenicity test results. Strains like TA1535 and TA100 are more sensitive to base-pair substitutions, while TA98 is better at detecting frameshift mutations. Strains carrying the pKM101 plasmid exhibit high sensitivity to mutagens that induce oxidative stress or single-strand breaks in DNA [15,56]. For example, in the study by Blair et al. [60], mutagenicity was not detected in strain TA100 but was observed in TA98, highlighting the specificity of the mutagenic response between different bacterial strains.

The Ara test, using strain BA13 (ΔuvrB, pKM101), is highly sensitive to oxidative stress [43]. The mutagenic effects of caffeic acid were stronger in the Ara test [43] than in the Ames test [41,68,69,79,80]. Caffeic acid may induce oxidative stress rather than direct DNA base-pair substitutions, making it more detectable in the Ara test than the Ames test. In the Ames test (TA100), caffeic acid was only mutagenic in the presence of Mn^2+^ or under nitrosation [41,69]. Similarly, caffeic acid induced DNA strand breaks only in the presence of Cu^2+^ [70]. This suggests that transition metal ions enhance oxidation of caffeic acid, leading to the generation of ROS, which can cause DNA damage. This may explain why caffeic acid alone does not exhibit strong mutagenic effects in Ames and DNA strand breaks assays.

The choice of cell lines varies, with some studies using metabolically competent human-derived cells (e.g., HepG2, THLE, Caco-2) while others rely on rodent-derived systems (e.g., V79, CHO). Differences in metabolic capacity can influence results, particularly for compounds requiring bioactivation. The comet assay results for HMF in HepG2 cells reported by Severin et al. [94] contradict the findings of Janzowski et al. [93], who observed no DNA damage with HMF at concentrations up to 80 mM in V79 and Caco-2 cells using the same assay. HepG2 cells can metabolize HMF into sulfoxymethylfurfural (SMF), a potentially harmful metabolite, through the activity of the SULT1A1 sulfotransferase enzyme, which has been confirmed to be present in these cells. Glatt et al. [96] also demonstrated that V79 cells expressing SULT1A1 exhibited a genotoxic response to HMF across all tested concentrations (19.8–3808 µM), highlighting the role of SULT1A1 in HMF metabolism and its potential genotoxicity. In contrast, Caco-2 and V79 cells either lack SULT1A1 or exhibit very low enzymatic activity, which may explain why Janzowski et al. [93] did not observe genotoxic effects in these cell lines. The presence or absence of 3′-phosphoadenosine-5′-phosphosulfate (PAPS), an essential cofactor for sulfotransferase (SULT) activity, may also contribute to the discrepancies between positive and negative in vitro results for HMF [21].

In addition, different mutagenicity assays have differing sensitivity to different compounds, and variability in test endpoints can significantly influence the results [43]. Stich et al. [69] used multiple genotoxicity assays to assess the genotoxicity of chlorogenic acid and caffeic acid. In the Ames test, no mutagenic activity was observed, whereas other tests indicated genotoxic effects. Similarly, caffeic acid demonstrated genotoxicity in the chromosomal aberration test, which detects chromosomal breaks, gaps, and translocations [68,69]. However, in the micronucleus test, it did not increase micronuclei frequency or induce DNA tail formation in the comet assay [71]. Instant coffee also showed no genotoxic effects in the micronucleus and comet assays [25,55]; however, it exhibited genotoxicity in the Ames test [18,19]. Similar findings were reported by Da Silva et al. [39], Reddemann [82], Fung et al. [68], Severin et al. [94] and Janzowski et al. [93], where different assays yielded varying results. Conflicting results were also observed in the Ara test [43] and the Ames test [56] for coffee grounds. In a study by Severin et al. [94], although HMF induced DNA strand breaks in the comet assay, no micronuclei formation was observed. The authors suggest that the DNA breaks detected in the comet assay may have been efficiently repaired, preventing micronucleus formation.

Some assays exclude metabolic activation (S9 fraction), which may underestimate the mutagenic potential of coffee constituents that require enzymatic conversion to reactive metabolites. For example, in studies by Johansson et al. [46] and Da Silva et al. [39], roasted coffee and instant coffee exhibited genotoxicity in Ames test only with S9. HMF is also generally considered mutagenic only after metabolic activation by sulfotransferases (SULTs) [94,96], suggesting that its mutagenicity is metabolism-dependent.

Differences in coffee preparation (filter, espresso, instant, purified instant coffee), coffee bean source, species, variety processing and roasting conditions may lead to variations in chemical composition and influence in genotoxicity results. This is evident in the findings of Da Silva et al. [39] and Blair and Shibamoto [60], which evaluated the genotoxicity of coffee with different roasting degrees and different coffee preparations. Another example is the study by Liu et al. [40], which did not concentrate the coffee or use chemical extraction methods, and diluted Americano coffee samples showed no detectable mutagenicity in the Ames test. In contrast, coffee samples that are highly concentrated or subjected to high temperatures exhibited direct mutagenic activity [33,37]. In study of Kato et al. [47] mutagenicity was also observed only after purification of instant coffee. Suwa et al. [45] demonstrated that brewed coffee exhibited mutagenicity in *S. typhimurium* TA100 when using well-roasted coffee in their experiments. In contrast, other studies [53,54] reported little to no mutagenic effects but did not specify the roasting level of the coffee used. This suggests that roasting conditions may influence the mutagenic potential of coffee.

Carbonyl compounds such as glyoxal, methylglyoxal, and diacetyl have been identified as contributing factors to the mutagenic effects of coffee samples [43,83,84,86]. Pure trigonelline is not mutagenic [68], but its mutagenic activity has been reported after heating, as it can be converted into potentially mutagenic compounds during coffee processing [67]. However, several studies have reported that coffee samples were non-mutagenic [25,40,53,54,55,57,62]. The lower concentrations of these compounds in non-mutagenic samples may explain the discrepancy compared to studies that detected genotoxic effects [15,18,33,36,42,46]. Therefore, variations in coffee processing methods, which are continuously evolving, likely lead to differences in chemical composition. These factors may account for the inconsistent findings regarding the mutagenic potential of coffee samples.

The different extraction methods also likely explain the varying genotoxicity results. Fernandes et al. [64] utilized acidic, alkaline, and prolonged water extraction for coffee waste, which may have enhanced the release of mutagenic compounds, leading to positive genotoxicity results. In contrast, Bravo et al. [65] and Monente et al. [62] employed water extraction from spent coffee grounds after brewing, a process that likely excluded many nonsoluble mutagens, resulting in negative genotoxicity findings.

In addition, differences in dosages and exposure times may explain the discrepancy in genotoxicity results. The dose ranges used across studies are inconsistent, with some exceeding physiologically relevant concentrations, potentially leading to false-positive results due to cytotoxicity rather than true genotoxic effects [53,54]. For example, in a study by Duarte et al. [41], the concentration of instant coffee tested in the Ames assay (0.1–0.8 mg/plate) was lower than in other studies (2–60 mg/plate) that reported genotoxic effects [18,19,33,45,48,51]. A similar pattern was observed for the genotoxicity of HMF in the comet assay [93,95], where differences in tested concentrations influenced the results.

In the yeast gene conversion assay, chlorogenic acid exhibited convertogenic activity at concentrations of 20, 40, and 80 mg/mL [69], whereas at a lower concentration (1 mg/mL), it did not induce mitotic conversion [76]. Similarly, chlorogenic acid at concentrations of 0.001–0.5 mM did not induce DNA damage in the comet assay [35,71]. However, at higher concentrations (0.5–5 mM) and a long exposure time (24 h), it caused DNA damage in cell lines [20]. Jiang et al. [104] found that CGA effectively scavenged superoxide and hydroxyl radicals at low concentrations but acted as a pro-oxidant at higher levels, leading to the formation of large DNA fragments and nuclear condensation in tumor cell lines [105]. Therefore, at high doses, chlorogenic acids may exhibit pro-oxidant effects, while at low doses, they can counteract oxidative stress and reduce DNA damage.

Mutagenicity studies on HMF have yielded conflicting results. In almost all studies, HMF showed no mutagenic effects in the Ames test [87,94,96,97], except for a study by Kim et al. [84], which tested only a single concentration and exclusively on strain TA100. However, at high concentrations, HMF exhibited genotoxic effects in the umu and comet assay [93].

Some reports test only a single or limited range of concentrations (Appendix A Table A1), making it difficult to establish a dose-dependent trend. The dose–response relationship is key to distinguishing true genotoxic effects from artifacts due to excessive concentration or cellular stress. Furthermore, cytotoxicity can mimic genotoxicity, particularly at high concentrations. While some studies conducted cytotoxicity assessments to establish appropriate concentrations for genotoxicity assays and reduce the risk of false positives [53,54,94], there are studies that did not evaluate cytotoxicity or report cell viability thresholds [18,38,84].

Although caffeine is a major coffee constituent, caffeine was not the main factor responsible for mutagenicity (Figure 2) [18,41,42,43,44,54]. However, findings on coffee’s effects on DNA repair remain inconsistent. While some studies reported **no significant impact of coffee on DNA repair enzyme expression in mice,** caffeine is recognized as a **DNA damage repair inhibitor** [106,107,108]. In animal models, caffeine at high dose significantly enhances chromosomal damage. Based on in vitro studies, caffeine does not appear to be genotoxic [109]. One mechanism by which caffeine increases cellular sensitivity to DNA damage is by disrupting repair pathways, particularly homologous recombination (HR) and nucleotide excision repair (NER) [108]. In addition, caffeine alone at high doses can induce intracellular oxidative stress [110], inhibit normal DNA replication, alter the rate of DNA synthesis, delay the entry of cells from G1 into S phase, and appears to release the DNA damage-imposed G2 block, allowing cells to enter mitosis prematurely. This release of cells from the G2 block often results in premature condensing of chromosomes (PCC), nuclear fragmentation, and chromosomal shattering [109]. Trigonelline may become mutagenic upon heating, especially in combination with amino acids [67,111]. As HHQ is linked to benzene metabolism, its presence in coffee raises concerns about potential genotoxicity [66]. HMF generally tested negative in genotoxicity assays but was mutagenic when metabolized by sulfotransferases (SULTs) from rats or humans [93,97,100,101]. Overall, HMF-induced genetic damage was found to be weaker than other foodborne genotoxins like acrylamide and furan [95]. Methylglyoxal and H_2_O_2_ are believed to account for the majority of the mutagenic activity observed in coffee [43,52,89]. Methylglyoxal induces oxidative stress, leading to increased ROS levels and activation of the p38MAPK, JNK, and NF-κB pathways, which regulate cell stress responses, inflammation, and apoptosis. Additionally, it triggers the ATM/Chk2/p53 DNA damage response pathway which plays a crucial role in detecting DNA damage [112]. H_2_O_2_ not only causes DNA damage but also inhibits DNA repair processes [113]. Caffeic and chlorogenic acids demonstrated different mutagenic responses in bacterial and mammalian systems depending on metabolic activation, suggesting a comprehensive genotoxicity assessment requires multiple test systems, as different systems can yield varying results due to differences in metabolism and DNA repair [68]. Metabolic activation also plays an important role in modulating the mutagenicity of compounds, emphasizing the need for metabolic considerations in food safety evaluations. Additionally, chlorogenic acid has shown genotoxic potential under specific conditions, particularly in the presence of transition metals like Mn^2+^, emphasizing the need for comprehensive assessments of its safety in dietary contexts [69]. Chlorogenic acid’s ability to induce DNA damage and topoisomerase-DNA complexes raises concerns about potential long-term carcinogenic effects [20].

The implications of these findings are not limited to individual health; they also extend to environmental and industrial considerations. For instance, the release of coffee waste into the environment without adequate treatment could pose risks to aquatic ecosystems and soil health due to the potential genotoxicity of leachates [64]. Future research should focus on identifying and characterizing novel mutagens generated during coffee processing, as well as their bioavailability in humans. Further study is also needed to determine the long-term effects of coffee use on genomic stability.

While coffee and coffee by-products demonstrate mutagenic potential primarily through H_2_O_2_ generation and the induction of oxidative stress, they also possess antioxidant properties that counteract oxidative damage [114]. Coffee contains bioactive antioxidants, including melanoidins, polyphenols (chlorogenic acid, caffeic acid), diterpenes (kahweol and cafestol), and caffeine, which play a crucial role in neutralizing free radicals and protecting DNA from oxidative damage [115]. In vitro studies indicate that kahweol and cafestol reduce genotoxicity in human liver cancer cells by activating phase II detoxification enzymes, particularly glutathione *S*-transferase (GST) and UDP-glucuronosyltransferase, while simultaneously inhibiting carcinogen-activating CYP450 enzymes. Additionally, these diterpenes enhance the expression of DNA repair proteins, supporting the repair of alkylation-induced DNA damage and contributing to cellular protection against genetic instability [116]. Cell and animal studies also suggest that coffee enhances antioxidant defense mechanisms by inducing mRNA and protein expression of key antioxidant enzymes through the Nrf2/ARE (antioxidant response element) pathway [27,58]. Both light- and dark-roasted coffee influence the expression of glutathione *S*-transferase (GST), heme oxygenase 1 (HO-1), and NAD(P)H:quinone oxidoreductase 1 (NQO1), which play essential roles in detoxification and cellular protection. Additionally, a 4-week coffee intervention study in healthy volunteers demonstrated increased glutathione and glutathione reductase levels, further supporting coffee’s role in contributing to the body’s antioxidant defense system [117,118]. Hernandes et al. [71] suggest that chlorogenic acid may inhibit DNA methyltransferases, which could offer therapeutic benefits for blood cancers linked to DNA methylation issues. These results suggest that coffee’s protective effects extend beyond just detoxification and involve multiple mechanisms. Bichler et al. [59] suggested that coffee consumption might offer greater protection against oxidative DNA damage than diets rich in fruits and vegetables. In addition, DNA-protective effects of coffee consumption support epidemiological data linking coffee consumption to reduced risks of liver cirrhosis and hepatocellular carcinoma, potentially due to its ability to prevent oxidative DNA damage [59].

Overall, these findings indicate that while coffee consumption may be associated with DNA repair inhibition, checkpoint signaling disruption and oxidative stress induction, the overall genotoxic risk remains minimal due to its protective antioxidant and detoxifying properties. Coffee and its bioactive compounds can modulate Nrf2-dependent transcription, which regulates a diverse range of proteins involved in detoxification, antioxidant defense, protein degradation, and inflammation [119]. In this pathway, kahweol, chlorogenic acid, and *N*-methylpyridinium enhance Nrf2 nuclear translocation and transcription. Kahweol does so via PI3K and p38 signaling [120], while chlorogenic acid likely acts through PI3K [121]. *N*-Methylpyridinium increases both Nrf2 translocation and Nrf2 transcription (Figure 3) [122].

There are currently a lack of scientific data on the genotoxicity of some coffee by-products, including coffee flower, coffee parchment, coffee silverskin and coffee oil. To ensure their safety for human consumption, these by-products need to meet the novel food requirements, including comprehensive toxicological evaluations, particularly addressing potential genotoxicity—an essential component of chemical risk evaluation in food and feed safety. Further studies are necessary to assess their safety and support their potential applications in novel food products.

## 5. Conclusions

This review evaluated the genotoxic and antimutagenic potential of coffee, coffee by-products, and associated bioactive compounds based on published in vitro studies. Studies have yielded variable outcomes, ranging from observed mutagenic effects to no detectable genotoxicity. These inconsistencies reflect the chemical complexity of coffee as a mixture and underscore the influence of factors such as dosage, assay type, and metabolic activation on genotoxicity outcomes. Human epidemiological studies do not support a link between coffee consumption and increased cancer risk. On the contrary, strong evidence suggests protective effects, particularly a significant reduction in liver cancer risk among high coffee consumers. Inverse or null associations have also been reported for endometrial, colorectal, and prostate cancers, while no consistent links have been observed for other major cancer types [27]. Overall, the current evidence indicates that coffee and coffee by-products do not pose a significant carcinogenic risk and may offer protective benefits in some cases. However, with growing interest in the use of coffee by-products in novel food applications, further studies— including in vivo and human exposure assessments—are recommended to ensure long-term safety.

## Figures and Tables

**Figure 1 toxics-13-00409-f001:**
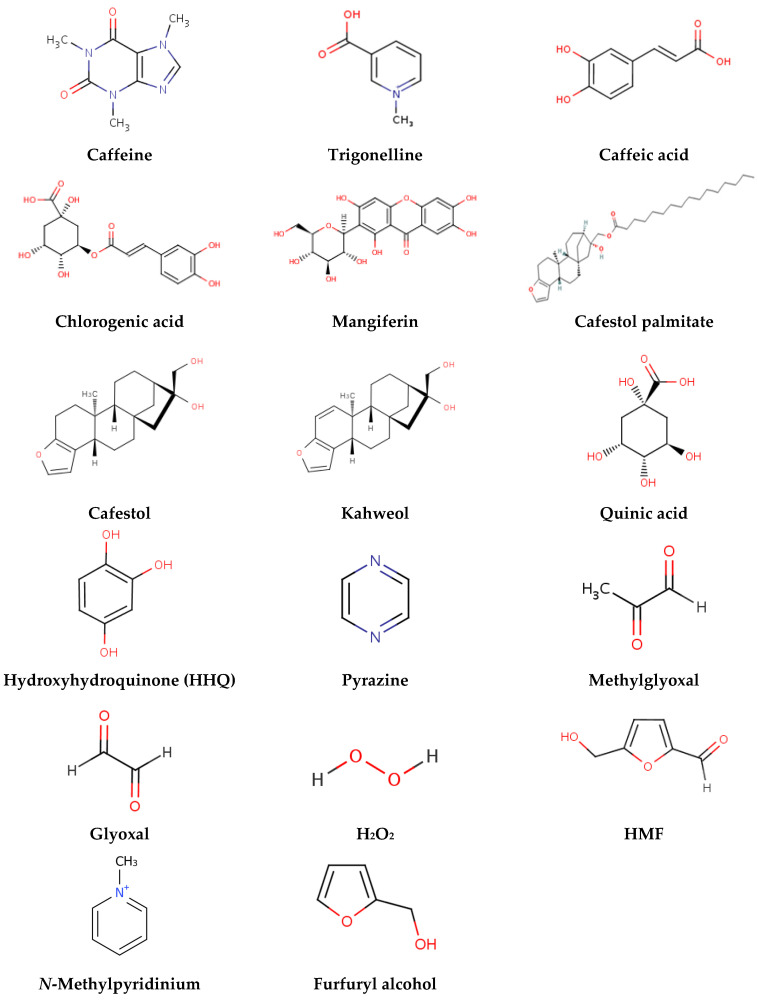
Chemical structures of key coffee constituents. Structures were obtained from ChEBI database, EMBL-EBI, Creative Commons License (CC BY 4.0).

**Figure 2 toxics-13-00409-f002:**
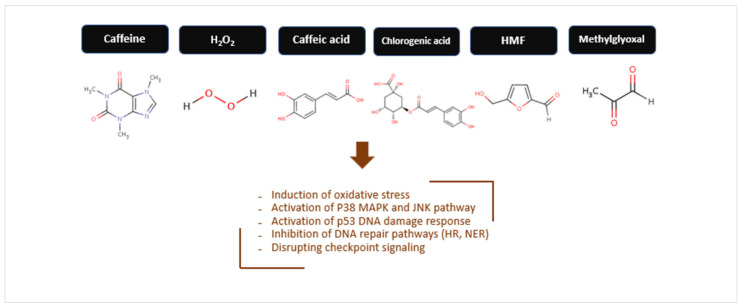
Potential in vitro genotoxic effects of coffee constituents.

**Figure 3 toxics-13-00409-f003:**
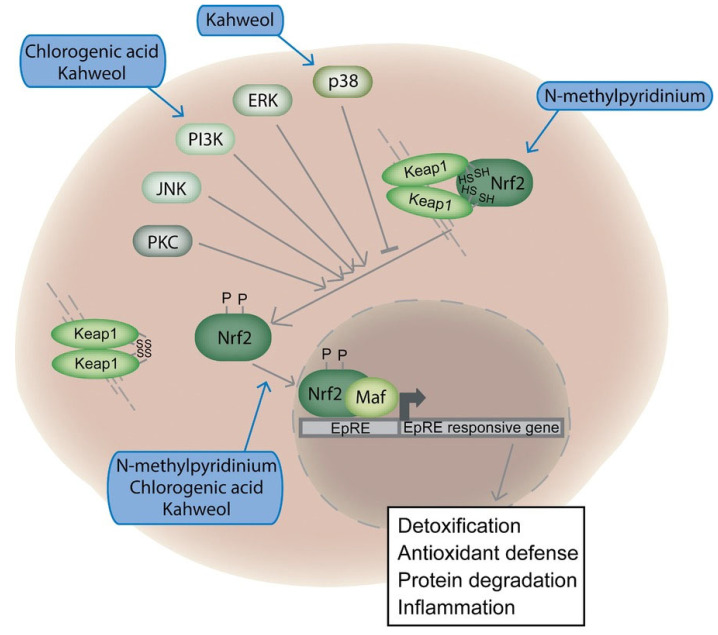
Molecular targets in the Nrf2 pathway for coffee compounds: chlorogenic acid, kahweol, and *N*-methylpyridinium. Reprinted with permission from Ref. [119]. Copyright 2013, John Wiley and Sons.

**Table 1 toxics-13-00409-t001:** Summary of in vitro genotoxicity and antimutagenicity studies on coffee, coffee by-products, and coffee bioactive compounds (details in Appendix A Table A1).

Test Material	Studies Showing Genotoxic Effects ^a^	Studies Showing No Effects ^a^	Studies Showing Antimutagenic Effects ^a^
Coffee cherry	–	[32]	–
Green coffee beans	–	[15,33,34,35,36]	[35]
Roasted coffee beans	[15,33,34,37,38,39]	[39,40]	–
Caffeinated instant coffee	[15,18,19,33,36,41,42,43,44,45,46,47,48,49,50,51,52]	[25,41,53,54,55]	[19,25,55]
Caffeine-free instant coffee	[15,18,36,42,46]	[25]	[25]
Caffeinated drip dark roasted coffee Decaffeinated drip coffee	[18]	–	–
Ground coffee	[43]	[56]	–
Caffeinated and decaffeinated coffee	–	[57]	[57,58]
Metal filtered coffee	[59]	[59]	[59]
Boiled coffee	–	[25]	[25]
Brewed coffee	[42,45,46,60,61]	[53,54,60,62]	
Silverskin extract	–	[63]	[63]
Spent coffee grounds	[64]	[62,65]	[62,65]
Water-insoluble fraction of coffee	–	[16]	[16]
HHQ	[66]	–	–
Trigonelline	[67]	[68]	–
Caffeic acid	[43,68,69,70]	[41,68,69,71]	–
Chlorogenic acid	[20,43,68,69,70,72,73,74,75,76,77,78]	[35,68,69,71,72,79,80,81]	[35]
Pyrazine	–	[68]	–
Mangiferin	[82]	[82]	–
Glyoxal	[43,83,84,85,86]	[87]	–
Methylglyoxal	[43,44,83,84,86]	–	–
Caffeine	–	[41,42,43,44,88]	–
H_2_O_2_	[43,52,89]	–	–
Coffee aroma	[42,90]	[90]	–
Cafestol palmitate	–	[91]	[91]
Cafestol and kahweol	–	[91]	[91,92]
Quinic acid	[69]	[69]	–
HMF	[84,85,93,94,95,96]	[87,93,94,97]	–
Furfuryl alcohol	[84,85,96]	–	–
Acetol	[84]	[87]	–
Diacetyl	[84]	[87]	–
Maillard reaction products	[98]	–	–

^a^ Assays used (Ames, comet, ara, umu, sister chromatid exchange assay, micronucleus, lac I test, chromosomal aberration, diphtheria toxin resistance, DNA-adduct formation, DNA breaking activity, DNA strand break assay, isolated λ DNA/plasmid DNA, mammalian cells mutagenesis, phosphorylated H2AX foci, gene conversion, PM2 bacteriophage DNA, HPRT and P53R assay); endpoints assessed (point mutations, single/double-strand breaks, forward mutations, chromosomal instability, micronuclei (breaks/loss), structural chromosomal damage, gene mutation, DNA adducts, p53 induction (apoptosis), loss of DNA integrity, gene recombination, modification in phage DNA, SOS-response induction); and key results (genotoxic activity observed, no genotoxic activity detected, antimutagenic).

## Data Availability

No new data were created or analyzed in this study. Data sharing is not applicable to this article.

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
