# Peer review of "Genotoxicity of Coffee, Coffee By-Products, and Coffee Bioactive Compounds: Contradictory Evidence from In Vitro Studies"

_toxics, 2025, doi:10.3390/toxics13050409_

Round 1
Reviewer 1 Report
Comments and Suggestions for Authors
The submitted review article aims to summarize the in vitro studies which analyzed the safety and potential toxicological profile of the coffee and its by-products. The authors are focusing on genotoxicity and mutagenic effect (anti or pro- ???). While the subject of the review could be important for the scientific community, the present manuscript requires clarification, elaboration, and refinement to enhance its quality and scientific depth.
The manuscript is too long and lacks clarity and should be restructured.
Comments:
1. The Introduction section presents in detail the EFSA recommendations for testing novel foods or ingredients. One phrase with references, and a short explanation should be sufficient.
2. Why write about fluorescence in situ hybridization (FISH) when there is not a study based on FISH results presented in the manuscript later on? And why explain the AMES test interpretation in the introduction section??
3. It is not clear the meaning of Table 1 "Summary of in vitro genotoxicity and antimutagenicity studies on coffee, coffee by-products, and coffee bioactive compounds". This table is just listing the references related with the subject??? Please add relevant scientific data on the table, such as: In Vitro Test Models Used, Genotoxicity and mutagenicity endpoints assessed, and so on...
4. Table A1 from appendix reflects in part the summarized data from manuscript. In my opinion this table should be in the main text of manuscript.
5. Please explain what positive mutagenicity effect of some compounds listed in Table 1A shows means. It is confusing to label a table as: "Results for genotoxicity and antimutagenicity of coffee, coffee by-products and coffee bioactive compounds" and then to present some of them as pro-mutagenic agents?!!? It is not clear what the authors want to emphasize in the title of the table.
6. Conclusion section is confusing. The authors only mentioned a couple of times that clinical studies have shown that coffee lacks carcinogenic potential. The manuscript is all about in vitro studies!!! The take-home message of this manuscript is missing. The authors should rewrite this section as well.
Author Response
Comments and Suggestions for Authors:
The submitted review article aims to summarize the in vitro studies which analyzed the safety and potential toxicological profile of the coffee and its by-products. The authors are focusing on genotoxicity and mutagenic effect (anti or pro- ???). While the subject of the review could be important for the scientific community, the present manuscript requires clarification, elaboration, and refinement to enhance its quality and scientific depth.
The manuscript is too long and lacks clarity and should be restructured. Comments:
- The Introduction section presents in detail the EFSA recommendations for testing novel foods or ingredients. One phrase with references, and a short explanation should be sufficient.
RESPONSE:
We have shortened the EFSA-related content in the Introduction to a concise sentence with a brief explanation and relevant references. The detailed description of EFSA’s tiered testing strategy has been relocated to the Results section under "Overview of Genotoxicity Testing Methods“, where it is more contextually relevant and better aligned with the subsequent review of genotoxicity studies on coffee and coffee by-products, which provides readers with essential background on testing methodologies before engaging with the study findings.
- Why write about fluorescence in situ hybridization (FISH) when there is not a study based on FISH results presented in the manuscript later on? And why explain the AMES test interpretation in the introduction section??
RESPONSE:
We have removed the mention of fluorescence in situ hybridization (FISH), as it is not directly relevant to the studies discussed in our manuscript. Additionally, the explanation of the Ames test interpretation has been relocated from the Introduction to the Results section under "Overview of Genotoxicity Testing Methods," where it more appropriately supports the context of the studies reviewed.
- It is not clear the meaning of Table 1 "Summary of in vitro genotoxicity and antimutagenicity studies on coffee, coffee by-products, and coffee bioactive compounds". This table is just listing the references related with the subject??? Please add relevant scientific data on the table, such as: In Vitro Test Models Used, Genotoxicity and mutagenicity endpoints assessed, and so on...
RESPONSE:
Table 1 was intended to categorize the available literature into three general groups—studies showing genotoxic effects, no observed effects, and antimutagenic effects—rather than summarize experimental outcomes or methodology. It serves as a reference overview to help readers navigate the scope of in vitro studies on coffee, coffee by-products, and related compounds. To avoid redundancy, detailed information such as test models, assay types and so on was instead presented in Table A1 of the Appendix. To address the reviewer’s concern and enhance clarity, we have revised the title of Table 1 to more accurately reflect its purpose and we included a brief note in the table footnote acknowledging the types of assays and endpoints assessed, while maintaining the focus on categorization rather than detailed data presentation.
- Table A1 from appendix reflects in part the summarized data from manuscript. In my opinion this table should be in the main text of manuscript.
RESPONSE:
We appreciate the reviewer’s suggestion. However, we believe that keeping Table A1 in the Appendix is more appropriate for maintaining the readability and flow of the main manuscript. Table A1 is extensive and contains detailed information on individual studies, including the assays types, experimental system and so on, which would significantly increase the length and complexity of the main text if included there. Instead, Table 1 in the main text presents a categorized summary of the key findings—"Genotoxic activity observed," "No genotoxic activity detected," and "Antimutagenic effect observed"—to guide readers through the overall trends in the referenced in vitro studies. We have now better cross-linked the tables (see changed header of table 1 and footnote).
- Please explain what positive mutagenicity effect of some compounds listed in Table 1A shows means. It is confusing to label a table as: "Results for genotoxicity and antimutagenicity of coffee, coffee by-products and coffee bioactive compounds" and then to present some of them as pro-mutagenic agents?!!? It is not clear what the authors want to emphasize in the title of the table.
RESPONSE:
We have revised the title of Table 1A to more clearly reflect its purpose. The updated title specifies that the table presents in vitro genotoxic and antimutagenic effects of coffee, coffee by-products, and their bioactive compounds, and includes brief definitions of positive and negative results to avoid ambiguity regarding the interpretation of the data.
- Conclusion section is confusing. The authors only mentioned a couple of times that clinical studies have shown that coffee lacks carcinogenic potential. The manuscript is all about in vitro studies!!! The take-home message of this manuscript is missing. The authors should rewrite this section as well.
RESPONSE:
We have rewritten the Conclusion section. The revised conclusion emphasizes the core findings and their relevance. The updated conclusion is now more aligned with the manuscript’s scope and purpose.
Reviewer 2 Report
Comments and Suggestions for Authors
This paper presents a comprehensive review of published articles on coffee and coffee by-products, focusing on their potential effects on humans. The authors identified 122 articles examining in vitro tests conducted to assess the impact of 38 diverse bioactive compounds on possible DNA damage. They systematically summarized and compared the results. While this review does not provide definitive conclusions regarding genotoxic effects, it represents a valuable contribution to the field. The article is thorough and well-structured, offering a detailed synthesis of existing research.
Author Response
Thank you for the assessment of our article!
Reviewer 3 Report
Comments and Suggestions for Authors
The topic of the review may be of interest to many readers.
The work is generally well written. The authors have prepared valuable data-rich appendix.
Some comments below:
- The paper mentions studies on the effect of caffeine on the risk of various cancers, e.g. liver, colorectal, prostate, breast, pancreatic, and bladder cancers, lung, esophageal, stomach, or ovarian cancer. There is no information in the manuscript about studies on the effect of caffeine consumption on melanoma risk, and such cohort studies have been published.
- Figure 3 - is taken from the publication [119]. Only a small modification has been made. In my opinion, the authors should have permission from the authors of the source publication to use their graphic.
- Figure 1 - the size of structures should be unified.
- Appendix requires edits to make the table data more readable.
- In vitro in vivo – no italic in some places
Author Response
The topic of the review may be of interest to many readers.
The work is generally well written. The authors have prepared valuable data-rich appendix.
Some comments below:
1. The paper mentions studies on the effect of caffeine on the risk of various cancers, e.g. liver, colorectal, prostate, breast, pancreatic, and bladder cancers, lung, esophageal, stomach, or ovarian cancer. There is no information in the manuscript about studies on the effect of caffeine consumption on melanoma risk, and such cohort studies have been published.
RESPONSE:
Thank you for your helpful comment. Our manuscript primarily focuses on in vitro genotoxicity and antimutagenicity studies of coffee, coffee by-products, and their bioactive compounds—rather than on the effects of individual constituents on specific cancer types. To ensure the conclusion section remains aligned with the manuscript’s core focus, we have rewritten it to emphasize the in vitro findings while briefly referencing human data for broader context. The mention of cancer risk associations (including liver and other cancers) in conclusion section is based on IARC evaluations, which we cited to support the point that, despite some compounds showing in vitro genotoxicity, current human data do not indicate a significant carcinogenic risk from coffee consumption. This reference is intended to underscore the need for further in vivo and human exposure studies, particularly as coffee by-products are increasingly considered for use in food and consumer products. For this reason, we have not expanded into additional cancer-specific data, such as melanoma, to maintain the manuscript’s focus and avoid overextending its scope.
2. Figure 3 - is taken from the publication [119]. Only a small modification has been made. In my opinion, the authors should have permission from the authors of the source publication to use their graphic.
RESPONSE:
We have obtained the necessary permission from the publisher and updated the figure caption accordingly to reflect this.
3. Figure 1 - the size of structures should be unified.
RESPONSE:
The updated figure has been included in the revised manuscript.
4. Appendix requires edits to make the table data more readable.
RESPONSE:
We acknowledge that the appendix table is quite extensive, which may affect its readability in the current format, which corresponds to the MDPI template. However, because the table is long and includes detailed information, it’s difficult to make major layout changes at this stage. We kindly suggest that final formatting improvements can be addressed by the journal’s copy editor during the production process to ensure optimal readability.
5. In vitro in vivo – no italic in some places
RESPONSE:
According to the journal's style guide, in vitro and in vivo should not be italicized. We have removed all italics for these terms throughout the manuscript to ensure consistency with the journal’s formatting requirements.
Round 2
Reviewer 1 Report
Comments and Suggestions for Authors
The manuscript was revised by authors. In this form could be published.